# Subsurface flow and phosphorus dynamics in beech forest hillslopes during sprinkling experiments: How fast is phosphorus replenished?

Michael Rinderer[1], Jaane Krüger[2], Friederike Lang[2], Heike Puhlmann[3], Markus Weiler[1]

[1] Chair of Hydrology, University of Freiburg, Freiburg, Germany
[2] Chair of Soil Ecology, University of Freiburg, Freiburg, Germany
[3] Department of Soil and Environment, Forest Research Institute Baden-Württemberg, Freiburg, Germany

*Correspondence to*: Michael Rinderer (michael.rinderer@hydrology.uni-freiburg.de)

**Abstract.** The Phosphorus (P) concentration of soil solution is of key importance for plant nutrition. During large rainfall events, the P concentration is altered by lateral and vertical subsurface storm flow (SSF) that facilitates P mobilization, redistribution within the soil profile and potential P export from the ecosystem. These processes are not well studied under field conditions. Important factors of the replenishment of P concentrations in soil solutions are the rate of P replenishment (by biotic and abiotic processes) and the P buffering capacity of soils. Lab experiments have shown that replenishment times can vary between minutes and months. The question remains how P concentrations in lateral and vertical SSF vary under natural field conditions. We present results of large-scale sprinkling experiments simulating 150 mm throughfall at 200 m$^2$ plots on hillslopes at three beech forests in Germany. We aimed at quantifying lateral and vertical SSF and associated P concentrations in the forest floor, the mineral soil and the saprolite during sprinkling experiments in spring and summer. The sites differed mainly in terms of soil depth, skeleton content and soil P stock (between 189 g/m$^2$ and 624 g/m$^2$ in the top 1 m soil depth). Vertical SSF in the mineral soil and in the saprolite was at least two orders of magnitude larger than lateral SSF in the same depth. Vertical and lateral SSF consisted mainly of pre-event water that was replaced by sprinkling water. Higher P concentrations in SSF in the first 1 to 2 h after onset of SSF indicated nutrient flushing, but P concentrations in the mineral soil and saprolite were nearly constant thereafter for most of the experiment despite strong increase in SSF. This suggests that P in the soil solution at all three sites was replenished fast by mineral or organic sources. If chemostatic transport conditions would dominate in SSF, annual P losses at the lateral and vertical boundary of a forest plot could be approximated by knowing the average P concentration and the water fluxes in forest soils. A rough estimation of the annual P loss based on this simplified assumption for one of our sites with longer SSF data, resulted in an annual P loss of 3.16 mg/m$^2$/a. This P loss is similar to estimates from a previous study at the same site using bi-weekly groundwater samples. Our approximated annual P loss in SSF was in a similar order of magnitude as P input by dry and wet deposition and by mineral weathering. Despite the fact that P losses from the ecosystem seem to be small, the translocation of P from the forest floor to the mineral soil might be of high relevance at sites with low P stocks where the forest floor is the dominant source for the P nutrition of trees.

**1 Introduction**

Phosphorus (P) is a major component of plant nutrition and has been reported to be a potential limiting factor of primary productivity in forest ecosystems (Achat et al., 2016; Elser et al., 2000, 2007). In the last decades a decrease of foliar P concentrations and an increase in the nitrogen to phosphorus (N:P) ratio has been observed in forests (Braun et al., 2010; Duquesnay et al., 2000; Jonard et al., 2015; Kabeya et al., 2017). From a plant nutrition perspective the replenishment of the P concentration of the soil solution is of key importance for the fertility of a soil (Lambers et al., 2008). The replenishment is a function of the exchange rate between the soil solution and the solid phase including biotic and abiotic processes. Helfenstein et al. (2018) present a worldwide compilation of P turnover times for 217 soils that range between $10^{-2}$ to $10^{6}$ minutes for which the majority have a turnover time of 1 to 100 minutes. The same authors also found a negative relation between the concentration of the soil solution P and the turnover time. In our paper we us the term P *replenishment* to describe the replenishment of P by both, biotic and abiotic processes. A high rate of replenishment implies that the soil has many potential binding sites where P can be sorbed or precipitated or the soil has many microorganisms that can immobilize or mobilize P. Thus, the P replenishment is also related to the P buffering capacity which is defined as the ability of a soil to moderate changes in the concentration of soil solution P (Pypers et al., 2006; White and Beckett, 1964).

The P dynamic in subsurface storm flow (SSF) is an indicator for dilution and enrichment processes along the lateral and vertical flow paths in the soil (Bol et al., 2016; Heathwaite and Dils, 2000; Julich et al., 2017a; Steegen et al., 2001). Unlike for agricultural land only very few studies exist that quantify the dynamics of P concentrations in forest ecosystems, as concentrations are low and measuring vertical and lateral SSF is challenging. In a small number of field studies water was sampled from surface waters i.e., forest streams (Cole and Rapp, 1981; Gottselig et al., 2017; Kunimatsu et al., 2001; Schindler and Nighswander, 1970; Tayor et al., 1971; Zhang et al., 2008) or from groundwater wells in riparian zones near the stream that are easier accessible than SSF (Carlyle and Hill, 2001; Fuchs et al., 2009; Vanek, 1993). But elemental composition of stream water represents an integrated signature of the entire catchment and is therefore not appropriate for a detailed process identification within soil compartments. Stream water is also subject to in-stream retention and mobilization of P and therefore not necessarily representative of transport conditions in hillslopes (Gregory, 1978; Hill et al., 2010; Mulholland and Hill, 1997; Sohrt et al., 2019; Stelzer et al., 2003).

Therefore, other studies collected soil solution below forest stands using suction lysimeters (Cole and Rapp, 1981; Compton and Cole, 1998; Kaiser et al., 2000; Qualls et al., 2002). A very limited number of studies used a trench to measure P concentration in lateral SSF. Timmons et al. (1977) were among the first; they installed a 1.8 m long drainage at the intersection of the A and B horizon in 33 cm soil depth to measure water fluxes and P concentrations in an aspen-birch forest in Minnesota. Sohrt et al. (2018) used 10 m wide trenches to analyze lateral subsurface flow from the forest floor, mineral soil and saprolite at three beech forest stands in Germany during natural rainfall events. Jackson et al. (2016) performed an artificial sprinkling experiments on a 200 m$^2$ plot on a hillslope at the Savannah River Research Site (South Carolina, USA) covered by pine trees and measured water flux and P concentrations in lateral SSF at 1.25 m soil depth. They applied dye- and conservative tracers

to identify dominant flow paths and fractions of event and pre-event water involved with solute transport. Makowski et al. (2020) performed artificial sprinkling experiments on 3 m² plots and reported significantly higher P concentrations in the sampled vertical SSF in the first two hours (P flushing), followed by decreasing and finally constant P concentrations for the rest of the experiment.

From field studies as described above and from lab experiments using soil columns we know that transport of P during high intensity rainfall events occurs mainly along preferential flow paths (Cox et al., 2000; Fuchs et al., 2009; Julich et al., 2017b; Missong et al., 2018a). Preferential flow paths allow subsurface flow to bypass the soil matrix that otherwise has been shown to effectively retain P (Compton and Cole, 1998; Ilg et al., 2009; Johnson et al., 2016; Qualls et al., 2002). Moreover, biopores (e.g., earthworm borrows, root channels) have been identified as biochemical hotspots that can show significantly higher P concentrations than the soil matrix, especially in fine textured soils (Backnäs et al., 2012; Hagedorn and Bundt, 2002). Preferential flow paths can extent below the rooting depth and are therefore considered an important pathway of P losses from forest ecosystems (Julich et al., 2017a; Sohrt et al., 2017).

Here we present a comparative study based on hillslope-scale artificial sprinkling experiments on 200 m² plots at three beech (*Fagus sylvatica*) forest sites that differ in terms of their soil depth, skeleton content and SSF flowpaths as well as their soil P stocks. We used an experimental setup that allowed to measure soil-depth specific lateral and vertical flow of water and transported hydrochemicals (incl. P). We performed two sprinkling experiments at each site to capture potential differences in P dynamics within the vegetation period (i.e., spring and summer/fall). The rational is that microbial activity, which is mainly responsible for P mineralization, is strongly dependent on moisture and temperature conditions (Brinson, 1977; Kirschbaum, 1995). In particular, we address the following research questions:

1. What are the main runoff generation mechanisms, flow paths and temporal delays in runoff response (lateral versus vertical, shallow versus deep) in three forest stands with different soil properties (i.e., soil texture, skeleton content, soil depth) during long, moderately intense sprinkling events?
2. What is the dynamic of P concentrations in lateral and vertical SSF, measured at different soil depths during artificial sprinkling events and does it differ seasonally and among sites with different physical soil properties and soil P availability?
3. Is P in the soil solution diluted during large sprinkling experiments or is the rate of P replenishment at all sites and all soil depths high enough to facilitate constant P concentrations (chemostatic conditions)?

We hypothesize that

1. in coarser textured soils vertical flow paths dominate and lead to faster and higher SSF at deeper depths than in fine textured soils.
2. P concentrations in SSF from the forest floor are high compared to the mineral soil and saprolite where P concentrations are lower due to adsorption processes. In summer the P concentrations in SSF are higher due to higher microbial activity and drought effects.

3. P concentrations in SSF are not diluted by the large amount of sprinkling water as P stocks are by far larger than P losses during a single event and as P replenishment is expected to be quick.

## 2. Methods

### 2.1 Study sites

For this analysis three beech (*Fagus sylvatica*) forest sites in Germany with contrasting soil hydrological properties (i.e., skeleton content and soil depth) and P stocks were selected. Their site characteristics are summarized in Tab. 1. Mitterfels (MIT) (48° 58' 32" N; 12° 52' 37" E) is located ca. 70 km east of Regensburg in the Bavarian Forest at 1023 m a.s.l. Its mean annual precipitation is 1299 mm. The site is characterized by Hyperdystric chromic, folic cambisol with a loamy topsoil (0 – 35 cm) and a sandy-loamy subsoil (35 – 130 cm). The stone content in the top- and subsoil is 23 % and 26 % and the P stock in the upper 1 m of the soil profile is 678 g/m$^2$. The saprolite reaches a total depth of 7 m but is less weathered below 2 m depth. The parent material below the saprolite is paragneiss. Conventwald (CON) (48° 01' 16" N; 7° 57' 56" E) is located 20 km east of Freiburg in the Black Forest at 840 m a.s.l. and has a mean annual rainfall of 1749 mm. The parent material, main soil type and vegetation is similar to MIT but soils considerably differ in the skeleton content (CON: 87 % topsoil, 67 % subsoil), the depth of the saprolite (CON 17 m but less weathered below 3 m) and the P stock in the upper 1 m of the soil profile (CON: 231 g/m$^2$). Tuttlingen (TUT) (47° 58' 42" N; 8° 44' 50" E) is located 125 km south of Stuttgart at 835 m a.s.l. and has 900 mm annual rainfall. Due to its carbonatic parent material a rendzic Leptosol with a clayey top- and subsoil has developed (P stock 209 g/m$^2$). The soil profile has a 20 - 40 cm deep topsoil and a 60 - 80 cm deep subsoil directly overlaying the fractured carbonate parent material. The stone content of the top- and subsoil is 50 % and 67 %, respectively. The site is also covered by beech but the stand is younger than at MIT and CON. Soil bulk density of CON and TUT is more similar than compared to MIT, but all three soil profiles show considerably variation in the bulk density with depth (Fig. 1).

### 2.2. Experimental setup and lab analysis

At each of the three sites we delineated an experimental plot of 200 m$^2$ (10 m by 20 m) which was separated from its uphill neighboring area by a plastic foil inserted into the soil profile. At the downhill side of the plots, a trench (TR) was dug down into the saprolite with a hydraulic shovel excavator, and drainage mats and drainage pipes were installed in three (MIT, CON) or two (TUT) depths (Fig. 2) to collect lateral flow. The actual depth of the pipes varied according to the site-specific soil profile (Tab. 2), but was chosen such that water draining from the forest floor (L, Of, Oh), the mineral soil (A and B horizon) and the saprolite (Cw) could be sampled. Plastic foil was installed across the entire 10 m width of the trench and down to the depth of the three soil compartments so that all water flowing laterally towards the trench was captured in the appropriate drainage pipe. To measure also vertical flow, we installed zero tension lysimeters (LY) for which we used steel piling plates with a dimension of 1.0 m by 0.6 m. To install them, a trench was dug at the side of the hillslope and the steel piling plates were pushed from the side into the undisturbed soil profile with heavy duty hydraulic jacks. By this, effects on soil structure

by excavation and refill were prevented and the mixing of soil P stocks from different soil horizons was avoided. We installed the LY in similar depths as the TR. At MIT and CON we installed an additional LY right below the A-horizon in 30 to 40 cm

soil depth; at TUT the shallow soil depth did not allow installing a LY below 60 cm. In the following, the TR and LY are numbed as TR1B to TR3B and LY1B to LY4B with increasing soil depth (Tab. 2). ("B" indicates the sprinkling plot and allows distinction from another dataset not used in this paper but mentioned in subsequent publications). All trenches were backfilled after installation.

In the upper and lower half of the hillslope, volumetric water content and soil temperature were monitored at two soil profiles

in 20, 40, 60, 80 and 120 cm soil depth (no 120 cm measurements at TUT due to shallower soil). We used SMT100 sensors (Truebner GmbH) attached to CR1000 data loggers (Campbell Scientific) and monitored volumetric water content and soil temperature in 5 min time intervals.

At each plot we performed two artificial sprinkling experiments – one in spring at the start of the growing season and one in late summer/early fall during or towards the end of the growing season but well before leaf senescence. The two periods were

chosen to reflect potential seasonal differences in soil moisture and P supply, i.e., higher soil moisture content after snowmelt but less P mineralization due to colder temperature in spring, versus drier soil moisture conditions and advanced P mineralization with warmer soil temperature in summer/fall. The mean difference of the median volumetric water content over the 7 days preceding the experiment was 8 vol %, 3 vol % and 8 vol % between summer and spring for MIT, CON and TUT, respectively. We sprinkled the 200 $m^2$ plots with a mean rainfall intensity of 10 to 20 mm/h over 10 to 12 h. Although this

reflects a large rainfall event with a return period of more than 100 years for all sites (DWD Climate Data Center, 2010), the chosen rainfall intensity allowed all irrigation water to infiltrate into the soil and did not generate surface runoff. 60'000 liters of water were trucked to the site and run through an industrial deionizer (VE-300 (6x50 Liter), AFT GmbH & Co.KG) to reduce the mineral content to low levels typically found in natural rainfall. In our case the sprinkling water had an electrical conductivity of around 20 μS/cm. The sprinkling water was stored in a large water pillow (60'000 L, Sturm Feuerschutz

GmbH) between 50 and 100 m above the sprinkling plot. The resulting hydrostatic gradient was sufficient to run the sprinkling without a pump. The six radial sprinklers (Xcel-wobbler and pressure regulator manufactured by Senninger) installed at a height of 2 m sprinkled 60 % of the total water onto the 200 $m^2$ plot and 40 % outside to reduce boundary effects with the otherwise dry surrounding area. 1 kg (first sprinkling experiment) and 2 kg (second experiment) of 99.96 atom% deuteriumoxid was added while filling the water pillow to elevate the natural background deuterium isotopic signature of the

sprinkling water by ca. 100 permille. Water samples before and after adding the deuterium and during sprinkling (collected with totalizers on the experimental plot) were collected to measure the background isotopic composition and to check for a constant isotopic label signature over the course of the experiment.

The subsurface flow (SSF) from the TR and LY was routed outside the hillslope via a pipe system to tipping buckets that recorded the flow volume of SSF in 5 min time intervals. The pipe system had been flushed via access tubes the day before

the experiments to guarantee function and cleanness. Over the course of the sprinkling experiment (ca. 10 to 12 hours) the SSF of all TR and LY was sampled every 30 minutes into 100 ml brown glass bottles using automatic samplers (custom made by

the Chair of Hydrology, University of Freiburg). The sampling was continued for 12 hours after the end of the sprinkling with a sampling interval of 2 h. The water samples were transported in cooling boxes to the lab directly after the experiment for subsequent hydrochemical and stable isotope analysis.

To measure total phosphorus concentrations (Ptot), 50 ml of the sample was digested by adding 0.5 ml 4.5 M sulfuric acid ($H_2SO_4$) and processed in an autoclave. Ptot was analyzed by the molybdenum blue photometric method based on DIN EN ISO 6878 (DIN, 2004) using a Unicam AquaMate photometer (Spectronic Unicam) with a 5 cm-cuvette at 700 nm. We determined the limit of quantitation for Ptot (0.009 mg/l,) and the limit of detection (0.004 mg/l) based on DIN 32645 with a significance bound of 99 % for the limit of quantitation and 77 % for the limit of detection (DIN, 2008). The remaining 50 ml

of each sample was filtered with a 0.45 µm cellulose filter (PERFECT-FLOW, WICOM) and used for analysis of $^{18}O$ and $^2H$ stable water isotopes using a Cavity Ring-Down L2130-i Isotopic Liquid Water Analyser (Picarro Inc.). Based on the background isotopic signature and the isotopic signature of the sprinkling water, event and pre-event water fractions were calculated using a simple two endmember mass balance approach, also called two-component isotope hydrograph separation (Sklash and Farvolden, 1979).

**2.3 Data analysis**

TR and LY volumetric measurements were scaled to 1 $m^2$ plot area and expressed as mm/h, which allows direct comparison with the amount of incoming sprinkling water. We determined the time lag between the start of the sprinkling experiment and the time when 20 % of the total rise in SSF had been reached or exceeded (called $t_{rise20}$ in SSF). In a similar way we determined the time lag between the start of the sprinkling event and the time when the event water fraction had reached 20 % (called $t_{rise20}$

in event water fraction). The threshold of 20 % has been chosen as clear indication of first response in the change of SSF and event water fraction. If $t_{rise20}$ in event water fraction is longer than $t_{rise20}$ in SSF this indicates that the flow celerity is faster than flow velocity. If $t_{rise20}$ in event water fraction is shorter than $t_{rise20}$ in SSF, this indicates preferential flow as velocity is faster than celerity.

We plotted P concentrations of vertical and lateral SSF as a function of time but also as boxplots to allow a better comparison

between the experiments. To investigate if the P concentrations of each flow component was higher during the first 2 h after onset of flow than during the remaining time of the experiments, we plotted our data separately for the two periods of each event. As some of the flow components had only few samples in the flushing period a statistical test of significance was not meaningful. This was also the reason why we did not further differentiate between flushing period and remaining part for the following analysis.

We plotted P concentrations as a function of SSF in log-log space to investigate whether P mobilization was able to keep up with P transport. In this case, we would expect near chemostatic conditions, i.e., P concentrations would not vary much with increasing or decreasing SSF. In this case data points were expected to plot parallel to the x-axis. In case of simple dilution, P concentrations would decrease proportional with increasing SSF and the data points would be aligned on a 1:-1 line. As a further measure we calculated the ratio between the range in observed SSF values and the respective range in observed P

concentration. Under predominantly chemostatic conditions, the range in SSF would be much larger than the range in P concentrations.

In addition, we investigated the relation between Ptot concentrations and event water fractions and checked whether the slope of a linear regression based on the datapoints was significantly different from zero using a two-sided t-test and an Alpha =0.05. We also tested if this regression slope was significantly different from a slope describing dilution due to proportional mixing of pre-event water with our sprinkling water using a two-sided t-test. The slope describing simple dilution was determined by a regression with the best linear fit through the data points and the additional constrain of Ptot = 0 mg/l for event water fraction = 1.

## 3. Results

### 3.1 Lateral versus vertical SSF

In general, vertical SSF (measured by LY) dominated total water flow during all sprinkling experiments. Depending on site and soil depth, between 89 % and 99 % of total SSF percolated vertically to deeper depths and only < 1 to 11 % of total SSF was flowing laterally towards the trench (Fig. 3). At all study sites LY1B (below the forest floor) yielded steady flow with a mean rate of 10 to 15 mm/h, which is identical to the sprinkling rate. This confirms that the LY, even if positioned at the boundary of the experimental plot, were experiencing rainfall intensities that are representative for the rest of the plot. The LY at deeper soil depth showed a slower increase in SSF than LY1B, but also reached a mean flow rate of 10 to 15 mm/h towards the end of the sprinkling experiment (except LY3B and LY4B at MIT in spring and summer and LY3B at TUT in summer) (Fig. 3). Lateral SSF (measured by TR) was at least two orders of magnitude lower than vertical SSF but increased constantly towards the end of the sprinkling experiments. An exception was TR2B at TUT which reached a plateau (ca. 1 mm/h) during the experiment in spring, most probable due to wetter antecedent conditions. Maximum lateral SSF from TR2B at CON and TUT was 0.72 and 1.09 mm/h during the experiment in spring and 0.51 and 0.70 mm/h in summer. TR3B at MIT yielded no SSF in any of the two sprinkling experiments. This is likely attributed to lower skeleton content and higher storage capacity of the soil in MIT.

### 3.2 Event and pre-event water fractions

Vertical SSF in the topsoil, subsoil and saprolite (i.e., all LY except LY1B) at all sites and during all experiments was dominated by pre-event water (Fig. 3, Tab. 3). In contrast, the mean pre-event water fraction from the forest floor (i.e., LY1B) during the sprinkling events was low (i.e., MIT: 12 % and 15 %; CON: 10 % and 16 % and TUT: 4 % and 4 % in spring and summer, respectively). The mean pre-event water fractions of vertical SSF increased with depth at all sites and all events and already in 35 to 40 cm soil depth (LY2B) the mean pre-event water fractions during the events in spring and summer were 83 % and 88 % at MIT, 60 % and 58 % at CON, and 64 % and 83 % at TUT. The mean pre-event water fraction in the vertical SSF further continued to increase with soil depth (see LY3B and LY4B in Tab. 3).

For the lateral SSF a similar increase of the pre-event water fraction with soil depth was observed. However, the mean pre-event water fractions of lateral SSF in the subsoil (i.e., TR2B) were typically smaller than the pre-event water fractions in vertical SSF at similar depth (i.e., LY3B). The mean pre-event water fractions of TR2B during the events in spring and summer were 44 % and 33 % at MIT, 63 % and 55 % at CON and 70 % and 47 % at TUT. The mean pre-event water fractions during the events in spring and summer of LY3B were 95 % and 93 % at MIT, 63 % and 83 % at CON and 95 % and 70 % at TUT. Mean pre-event water fractions of the lateral and vertical SSF in the saprolite (i.e., TR3B and LY4B) were similar (i.e., 78 % and 74 % for TR3B and 86 % and 78 % for LY4B at CON during the experiment in spring and summer. (No flow from TR3B at MIT and no TR3B and LY4B installed at TUT).

### 3.3 Differences in SSF and response time between spring and summer

In general, SSF differed less between the sprinkling experiments in spring and in summer than between the three sites. This was particularly true for the peak flows in SSF. An exception was TUT where peak flows for LY2B and LY3B were higher in spring compared to summer which is likely due to differences in antecedent wetness between spring and summer at TUT.

SSF between spring and summer differed mainly in respect to the response time of the individual TR and LY. This was most pronounced at MIT and least at CON (Fig. 3, Tab. 4). At MIT $t_{rise20}$ in SSF was on average 1.9 times longer in summer than in spring (except TR2B: 0.7 times shorter) (Tab. 4). At CON it was the opposite i.e., $t_{rise20}$ in SSF was on average 0.7 times shorter in summer than in spring; except for SSF in the saprolite (i.e., LY3B, LY4B, TR3B) for which $t_{rise20}$ in SSF was on average 1.3 times longer than in spring (Tab. 4). At TUT $t_{rise20}$ in SSF in the top and subsoil was 1.8 times longer in summer than in spring but $t_{rise20}$ in SSF in the forest floor (LY1B and TR1B) was 0.3 times shorter. At CON and TUT $t_{rise20}$ in SSF was not always increasing systematically with soil depth (e.g., LY3B responded earlier than LY2B at CON in spring and TUT in summer; TR3B responded earlier than TR2B at CON in spring and summer). At MIT in summer LY4B did not respond until 1.5 h after the end of the sprinkling experiment and TR3B did not respond at all in spring and summer.

The $t_{rise20}$ of the event water fraction (Tab. 5), was typically longer than $t_{rise20}$ of SSF (Tab. 4). Only for TR2B and TR3B at CON in spring and summer and TR2B at TUT in summer $t_{rise20}$ of event water fraction was shorter than $t_{rise20}$ of SSF. These differences in $t_{rise20}$ of event water fraction and $t_{ries20}$ of SSF for these TR was in the range of 3 hours to almost 4 hours for CON and 1 h for TUT. At MIT, SSF from LY3B and LY4B did not reach 20% event water fraction during the experiments in summer and spring. The same was true for LY3B at TUT in spring.

### 3.4 Dynamics of P concentrations

Median Ptot concentrations of vertical SSF (LY) during the first flush (i.e., the first 1 to 2 hours after the first response of each flow component) were up to 6 times higher than during the remaining time of the sprinkling experiment (Fig. 5); some of the Ptot concentrations of the very first water samples were even up to one order of magnitude higher (Fig. 4a). This was particularly true for the vertical flow from the forest floor and the topsoil (LY1B and LY2B) but also apparent, albeit less distinct, for the subsoil and the saprolite (LY3B, LY4B).

The Ptot concentrations of vertical flow from the forest floor was significantly higher (Mann-Whitney Test, Alpha = 0.05) than in the topsoil (in 30 to 40 cm soil depth), except for CON and TUT in summer (Fig. 4a). Similarly, the Ptot concentrations of vertical flow in the subsoil and saprolite were significantly lower than in the topsoil except for MIT in spring and summer and at TUT in summer. All Ptot concentrations were above the limit of quantitation (i.e., 0.009 mg/l).

The Ptot concentrations in the same vertical flow component were not significantly different between spring and summer, except for LY1B and LY3B at MIT; LY1B at CON; LY1B, LY2B and LY3B at TUT. However, the Ptot concentrations of the same flow component (e.g., LY1B) at MIT was generally significantly higher than at CON and TUT, except for LY2B in spring. In a similar vein, Ptot concentration at CON were significantly higher than at TUT except LY3B in summer.

Ptot concentrations in the lateral flows (TR) also showed a sharp decline during the first 1 to 2 hours after the onset of flow of each component, except for TR1B and TR2B at MIT in spring and TR2B at CON in spring (Fig. 4b). Ptot concentrations of TR2B at CON in summer showed a steady decline and for TR3B the decline occurred with more delay (i.e., 5 h after first response) compared to the other experiments. At CON and TUT the Ptot concentrations in the lateral flow of the forest floor (TR1B) were significantly higher than in the subsoil (TR2B). Contrarily at MIT the Ptot concentrations in lateral flow of the forest floor (TR1B) were significantly lower than in the subsoil (TR2B), both in spring and summer (Fig. 4b). The difference in Ptot concentrations in the same flow component in spring and summer was not significant, except for TR1B at TUT.

### 3.4.1 P Concentration as a function of instantaneous flow

The range in SSF was typically several factors if not orders of magnitude larger than the range in Ptot concentrations during the sprinkling experiments except for those flow components that yielded little SSF in general (e.g., TR1B at CON during spring and summer , LY3B and TR2B at MIT during summer) (Fig. 6). Data points were in general aligned rather parallel to the x-axis in the log-log plots of Ptot concentration versus SSF and not along the 1:-1 line (Fig. 6). This suggests that transport in most subsurface flow components was rather chemostatic than diluted. However, we observed weak anti-clockwise hysteresis effects i.e., median Ptot concentrations were higher on the rising limb than on the falling limb of the SSF hydrographs, except for TR1B at MIT in summer, LY4B at CON in spring and LY2B at TUT in summer. Most differences in Ptot concentrations during rising and falling limbs were, however, small (e.g., 0.007 mg/l, 0.028 mg/l and 0.081 mg/l for the 25%-, 50%- and 75% quantile of all differences) indicating that the hysteresis effect was small and only for LY1B this difference was consistently significant across all sites (Suppl. Tab. 1).

### 3.4.2 P concentration as a function of event water fraction

Ptot concentrations also did not change significantly with increasing event water fraction (Fig. 7). Only in the forest floor (LY1B and TR1B), the slopes of the linear regression lines fitted to the Ptot concentrations as a function of event water fractions were significantly different from zero, except LY1B at CON in summer and TR1B at MIT in spring (Tab. 6). The transport conditions in the lateral and vertical SSF in the forest floor were therefore predominantly non-chemostatic. In the mineral soil and saprolite lateral and vertical SSF was dominantly chemostatic (except LY2B and LY4B at MIT in spring,

LY3B at CON in summer, LY2B at CON in spring, LY2B and TR2B at TUT in summer and TR2B at TUT in spring) (Tab. 6). Most of these regression slopes were however close to zero.

For the regression slopes indicating non-chemostatic behavior, we tested if they were not significantly different from a slope describing proportional mixing. In general this was not the case, except for some flow components from the forest floor (i.e., LY1B at MIT and TUT in summer, TR1B at CON in spring; see italic values in Tab. 6) and from the mineral soil (i.e, TR2B

at MIT in summer, TR2B at TUT in summer and spring and LY3B at CON in summer). At least for the latter three the regression slope was however close to zero (Tab. 6). As a rough generalization one could summarize: The regression slopes of most lateral and vertical SSF in the mineral soil and saprolite tended to be small or close to zero (the majority was chemostatic) whereas regression slopes of flow components from the forest floor were typically significantly different from zero (some indicating proportional mixing).

## 4. Discussion

### 4.1 Main SSF flow paths during long, moderately intense sprinkling events

In general, vertical SSF dominated total SSF during all sprinkling experiments and lateral SSF was at least two orders of magnitude lower than vertical SSF (Fig. 3). This finding implies that previous studies at trenched hillslopes at sites with well drained soils and moderately permeable bedrock missed out to quantify the important loss term of the water balance (e.g.,

Jackson et al., 2016; Sohrt et al., 2018; Timmons et al., 1977). This is partly due to different research foci of these studies but mainly attributed to the technical difficulty to measure and sample vertical SSF. The use of our large zero tension lysimeters is a successful way to capture vertical flow in heterogeneous soils. They yield more representative results than traditional small size lysimeters of a few $cm^2$ or suction cups that are more likely to be affected by soil heterogeneity. The steel piling plates that we pressed into the undisturbed soil profile from the side of the hillslope also allow to preserve the natural soil profile

above with its soil texture and structure and horizon-specific P stocks. However, their installation comes with high technical and man-power effort.

Vertical and lateral SSF in the mineral soil and saprolite at all sites and all experiments was predominantly pre-event water (Fig. 3, Tab. 3). This is generally in agreement with Jackson et al. (2016) who performed a tracer experiment at the Savannah River Site (South Carolina, USA) in a loamy sand topsoil overlaying a sandy clay-loam subsoil. However, their maximum pre-

event water fraction was 50 % while it was typically higher in our study (mean pre-event water fraction in vertical and lateral SSF of the mineral soil was 83 % and 63 %, see also Tab. 3). Our findings suggest that SSF runoff generation was dominated by initiation of water already stored in the soil, i.e., incoming event water was pushing pre-event water down into the soil profile initiating SSF. A non-sequential onset of SSF with soil depth and shorter $t_{rise20}$ of event water fraction than $t_{rise20}$ of SSF in some lateral flow components at CON and TUT suggest, however, that preferential flow occurred in parallel to matrix flow

at CON and TUT.

Occurrence of preferential flow is important as it allows P bypassing the soil matrix in its soluble and colloidal form and is therefore considered to be a very prominent pathway of P loss from the ecosystem (Jardine et al., 1990; Kaiser et al., 2000; Missong et al., 2018b). The fact that we observed indications of preferential flow predominantly at CON and TUT but not at MIT and mainly in lateral flow and less in vertical flow may be explained by differences in soil properties, especially skeleton content (Tab. 1) and soil bulk density (Fig. 1) of the three sites. Swelling and shrinking due to the higher clay content and more biopores as a result of higher earthworm abundance (due to higher soil pH) at TUT might be another mechanism leading to preferential flowpaths. At MIT the lateral and vertical flow components in the saprolite did not respond at all or with strong delay to the sprinkling experiments (Fig. 3, Tab. 4) which suggests very efficient storage of the sprinkling water in the soil. Therefore, at MIT characterized by relatively low skeleton content but high soil storage capacity, the shallow flowpaths were most important and the deeper flowpaths did not yield much or any flow. At CON the opposite was the case. The higher skeleton content allowed water to reach deeper soil depth and therefore deeper flowpaths yielded more flow than shallower ones. At TUT the clay-rich topsoil led to more lateral flow at shallow depth than CON but as the total soil depth was much smaller than at the two other sites, the storage capacity was less and therefore also the mineral soil yielded significant amounts of flow.

The differences in SSF between the sprinkling experiment in spring and summer were smaller than the difference in SSF among the three sites and mainly related to SSF response timing (Fig. 3, Tab. 4). The reason is likely due to relatively small differences in antecedent soil moisture conditions between the two sprinkling experiments in summer and spring (except TUT). This was particularly true at CON where the high skeleton content allows the soils to drain quickly to field capacity. Soil properties, such as drainable porosity, likely also explain why seasonal differences in SSF response timing were more pronounced at MIT than CON (Tab. 4, Tab. 5). At TUT the difference in SSF dynamics between the experiment in summer and spring is likely more related to differences in antecedent wetness conditions. The 7-day-average of the median volumetric water content of the soil profile at TUT during spring was 23 vol % compared to 15 vol % at TUT during summer. Still, the dominance of pre-event water fractions during both events at TUT suggest that not fast preferential flow but piston flow was the dominant process during both experiments at TUT.

**4.2 P concentration dynamics in vertical and lateral SSF**

Ptot concentrations in vertical and lateral SSF, particularly in the forest floor and to a lesser degree in the mineral soil, were typically significantly higher during the first 1 to 2 hours after the first flow response of each component than during the remaining time of the sprinkling experiment (Fig. 4, Fig. 5). This so-called nutrient flushing effect (Hornberger et al., 1994) has been described also in other studies as a prominent feature of lateral export of nutrients; mainly for N (DON) and C (DOC) (Qualls and Haines, 1991; van Verseveld et al., 2008; Weiler and Mcdonnell, 2006) but also for P (DOP) (Burns et al., 1998; Missong et al., 2018a; Qualls et al., 2002; Sohrt et al., 2018). Makowski et al. (2020), who measured P concentrations in vertical SSF, also reported nutrient-flushing in the first 2 hours of their sprinkling experiments. Various processes might contribute to this flushing effect. Drying and rewetting processes are known to induce nutrient flushing due to cell lysis due to

osmotic stress and subsequent release of nutrients (Gordon et al., 2008; Turner et al., 2003) as well as due to disruption of soil aggregates allowing mineralization of former inaccessible organic substrates (Bünemann et al., 2013). Furthermore, there is evidence that with rewetting after intense drought events the microbial activity is increasing. Phosphate mobilization has been observed in forest soils as a consequence (Brödlin et al., 2019). In addition, drying of organic rich material leads to water repellence of the forest floor material and thus might enhance particle-bound nutrient export (Dincher and Calvaruso, 2020).

In our study Ptot concentrations in SSF from the forest floor continued to decline after the flushing phase or were relatively constant (Fig. 4). The slope of a regression line of Ptot concentration versus event water fraction of most flow components in the forest floor was significantly different from simple dilution that would be the case if a given amount of P was leached and diluted by an increasing amount of event water in SSF (Fig. 7, Tab. 6). A likely better explanation could be that the rate of P replenishment in the P-rich forest floor was not high enough to facilitate the high P concentrations measured at the beginning of the event. This is in line with Helfenstein et al. (2018) who found a negative relation between P concentration in the soil solution and the turnover rate based on isotopic exchange kinetic experiments on 217 soil samples collected worldwide. The P concentrations from the forest floor (except MIT during summer) were more constant towards the end of the experiment, when the P concentrations from the forest floor where low in general.

In the mineral soil the flushing effect was less distinct. The P concentrations in SSF were generally lower compared to the forest floor. The change in Ptot concentrations in the mineral soil were several times smaller than the change in SSF (Fig. 6) and the regression slope of Ptot concentration versus SSF was not significantly different from zero for most flow components in the mineral soil (Tab. 6). This suggests that P transport in SSF in the mineral soil was chemostatic. It also suggests that the rate of P replenishment in SSF of the mineral soil was high during the experiments. This was even true towards the end of the sprinkling experiments when the fraction of new water increased. For this new sprinkling water, it is clear that the contact time was short (i.e., in the order of minutes to hours) and still the P concentration was not significantly lower than for water samples with a higher pre-event water fraction. The relatively constant P concentrations in the mineral soil also suggest, that P leached from the forest floor was efficiently buffered in the mineral soil and that this buffering effect is also apparent in soils with preferential flow paths (CON and TUT) that we considered as a potential bypass of the soil matrix. We could not see significant differences between the experiments in spring and summer at the three sites. An exception were the P concentrations of vertical flow from the forest floor at all sites.

### 4.3 Estimates of annual P fluxes

If one would argue that the amount of SSF during the flushing period is small compared to the remaining part of the event (see Fig. 3) one could also assume that chemotactic conditions prevail during large rainfall events. Given these simplifications, one can roughly estimate annual P losses from forest stands by knowing the amount of annual SSF and an average P concentration. This would be particularly relevant for long-term soil evolution modeling (Chadwick et al., 1999; Vitousek, 2004). We performed this hypothetical test based on the stated assumptions and estimated the annual P fluxes in SSF at CON, which is the only site where we have SSF data for more than 1 year. We chose the period between 10.05.2018 to 09.05.2019 in order

to not include the sprinkling experiments in the water balance. The sum of SSF and the calculated Ptot flux for this period is shown in Tab. 7. The mean P concentration used to calculate the annual P flux is based on the P concentrations measured after the first flush during the experiment at CON in spring 2018 (i.e., 07.05.2018). This is the sprinkling experiment most recent to the one-year period of interest. Our estimation of annual Ptot fluxes (sum of vertical and lateral) yielded 45 mg/m$^2$/a from the forest floor and 14 mg/m$^2$/a from the topsoil, 6 mg/m$^2$/a from the subsoil and 2 mg/m$^2$/a from the saprolite. Considering only the outside boundaries of the hillslope (LY4B, TR1B, TR2B, TR3B) we estimated a total P-loss of 3.2 mg/m$^2$/a. While we expect these values to be different for every year, the order of magnitude can be compared to values presented by Sohrt et al. (2019) who have sampled P concentrations in lateral SSF at CON on a bi-weekly interval between 01.03.2015 and 25.02.2016. Their mean P concentration from the organic layer (only lateral flow) was 0.57 mg/l, the annual water flux from the organic layer was 0.002 mm/a and the calculated annual P flux was 0.001 mg/m$^2$/a. For the mineral soil they reported a mean P concentration of 0.043 mg/l, a total annual water flux of 446 mm/a and a resulting P flux of 20 mg/m$^2$/a. While the annual flux from the mineral soil matches well between Sohrt et al. (2019) and our study, the annual P flux from the forest floor differs by several orders of magnitude. This is likely due to differences in the estimated water fluxes (Tab. 7). Sohrt et al. (2019) estimated the annual water flux by End Member Mixing Analysis (EMMA) based on samples from the lateral flow components (TR1B) only. Our annual water flux from the forest floor comprises both, the lateral (TR1B) and the vertical (LY1B) flow component of which the latter has been shown to be orders of magnitude lager then the lateral flow (see Fig. 3). For that reason it is likely better to compare the estimated P fluxes from the outside boundaries of the hillslope in our study (3.2 mg/m$^2$/a) to the P fluxes in the groundwater (2.5 mg/m$^2$/a) in the study of Sohrt et al. (2019). These two P fluxes represent the P loss from the entire hillslope. These two fluxes match well.

Our annual P fluxes are also in line with the range of fluxes published in a number of other studies or reviews. Bol et al. (2016) estimated that potential total P loss through leaching into subsoils and export by forest streams was less than 50 mg P/m$^2$/a. But the range of published data varies more than one order of magnitude (e.g., 4 mg Ptot/m$^2$/a in Benning et al. (2012); 32 mg/m$^2$/a in Julich personal communication, cit. in Bol et al. (2016)). Other studies reported only annual fluxes of Dissolved Organic Phosphorus (DOP). While these numbers are difficult to be directly compared to our Ptot data, the published values are in a similar order of magnitude (e.g.: 15 to 62 mg DOP/m$^2$/a in leachate from the organic layer and 1.7 to 38 mg DOP/m$^2$/a from the mineral soil (Kaiser et al., 2000, 2003; Qualls et al., 2000). In another meta-analysis Sohrt et al. (2017) presented annual P fluxes that are higher (e.g., mean: 226 mg/m$^2$/a (sd: 389 mg/m$^2$/a) and mean: 119 mg/m$^2$/a (sd: 279 mg/m$^2$/a) for water samples from the organic layers and the mineral soils). A possible reason for the higher values compared to our results could be that we used mean P concentration measured after the first flush which results in rather conservative estimates. Julich (personal communication cit. in Boll et al. (2016)) reported that up to 40 % of the annual P flux might occur during single events, which would suggest that the fluxes during the first flush have an important share on the annual flux and cannot be neglected. Further research is needed to clarify the role of P-flushing on the total P loss from forest ecosystems. Given the high P concentrations during the flushing phase of our sprinkling events it might be that relatively small amounts of SSF during the first flush can still export a significant amount of P and might be highly relevant especially at sites with low P stocks and/or P

availability (e.g., acid sites developed from P poor parent material (CON) or sites developed on carbonate rock (TUT)). The forest floor at such sites serves as most important P source and efficient P recycling from the forest floor is the prerequisite for a sufficient P-supply to the ecosystems (Hauenstein et al. 2018, Lang et al. 2017). In the mineral soil P is fixed to the solid phase by sorption to sesquioxides or precipitation of Ca-phosphates and therefore P availability is very low.

Despite the considerable range of published annual P fluxes in SSF from different forest environments it becomes clear that annual P fluxes are orders of magnitude lower than soil P stocks (Achat et al., 2016; Hou et al., 2018). In the case of CON soil P stocks are estimated to be 13 $g/m^2$ and 230 $g/m^2$ for the forest floor and the mineral soil, respectively (Lang et al., 2017). Therefore, export of P from forest ecosystems by SSF becomes only relevant on the time-scale of millennia (Bol et al., 2016; Vitousek, 2004). The annual P losses are also well compensated by atmospheric deposition and mineral weathering (Aciego

et al., 2017; Hartmann et al., 2014; Tipping et al., 2014). For instance, at CON the annual P fluxes by dry deposition, canopy leaching, bulk throughfall and mineral weathering are 10, 43, 60 and 76 $mg/m^2/a$ (Sohrt et al., 2017; Uhlig and von Blanckenburg, 2019) and therefore in a similar order as the estimated annual P fluxes by SSF in our study. It is also important to note that the internal ecosystem P fluxes due to forest litter fall (100 to 500 $mg/m^2/a$, Sohrt et al. (2017)) are up to one order of magnitude larger than the annual P fluxes in SSF. Tthis is an indication of an efficient nutrient recycling in forest ecosystems

(Lang et al., 2017).

## 5. Conclusions

We present results of sprinkling experiments conducted at 200 $m^2$ hillslopes at three beech forest sites in Germany that differ in their soil depth, skeleton content and soil P stocks, to quantify the dynamic of vertical and lateral SSF and associated P concentrations. Vertical SSF in the mineral soil and saprolite was at least two orders of magnitude larger than lateral SSF and

440 consisted mainly of pre-event water that was likely replaced by sprinkling water (piston flow mechanism). This suggests that previous studies that only measured lateral flow have likely missed the major hydrological export flux from forest stands. Median Ptot concentrations in SSF from the forest floor, and to a lesser extent also at deeper soil depths, at all sites were up to 6 times higher in the first 1 to 2 h after the first response of each flow component (first flush). For the remaining time of the experiments (ca. 10 h), transport conditions in the mineral soil and saprolite were however close to chemostatic. This suggests

that the rate of P replenishment at all three sites was high (in the order of minutes to hours). Our finding that chemostatic transport conditions were prevalent in the mineral soil for most of the duration of the experiments would suggest that annual P flux from forest stands could be approximated by simply knowing the average P concentration of the soil solution and the water balance of the site. A test of this assumption in the form of a rough approximation of the annual P flux at one of our sites (CON) yielded comparable results to an earlier study at the same site (Sohrt et al., 2019) and some other sites (Benning et al.,

2012; Bol et al., 2016). In terms of a P budget, our approximated annual P fluxes in SSF at CON were in a similar order as P inputs by dry and wet deposition and mineral weathering and several orders of magnitude smaller than the total P stocks of the

mineral soil. Despite the fact that P losses from the ecosystem seem to be small, the translocation of P from the forest floor to the mineral soil might be of high relevance at sites where the forest floor is the dominant source for the P nutrition of trees.

## 6. Data availability

Data is available under: https://1drv.ms/u/s!AhSOi7EfJ5qmfyJQrT3O65OLBuE?e=BrwfFp

## 7. Author contribution

MW wrote the grant; MR was responsible for the experimental design and field installations together with two technicians. MR planed and organized the sprinkling experiments and lab-analysis together with two technicians. MR was responsible for data pre-processing and all analysis. JK and FL provided the data on soil characterization and valuable thoughts and discussion 460 on the soil ecological aspect of the study. HP provided meteorological data, supported the long-term monitoring of SSF at CON and provided valuable feedback on the paper manuscript. MR wrote the manuscript including all figures and tables and all other co-authors discussed the results, provided valuable feedback on the text and figures.

## 8. Competing interests:

The authors declare that they have no conflict of interest.

## 9. Acknowledgments

This project was carried out in the framework of the priority program SPP 1685 "Ecosystem Nutrition: Forest Strategies for limited Phosphorus Resources" funded by DFG (grant No. WE 4598/7-2). The article processing charge was funded by the Baden-Wuerttemberg Ministry of Science, Research and Art and the University of Freiburg.

Special thanks to Jakob Sohrt for parts of the field installations (TR) and very insightful discussion and helpful suggestions 470 during the project. Many thanks to Emil Blattmann, Dominc Demand, Sandra Fritzsche, Maja Gensow, Nina Gottselig, Benjamin Grahler, Janine Heizmann, Barbara Herbstritt, Johannes Herre, Jakub Jeřábek, Britta Kattenstroth, Lara Kirn, Petra Küfner, Hannes Leister, Marvin Lorff, Anna Missong, Regine Nitz, Felix Nyarko, Benjamin Schima, Jonas Schwarz, Stefan Seeger, Jürgen Strub, Angela Thiemann, Delon Wagner, Monika Wirth-Lederer for support in the field, lab and admin. We also want to thank the team of the FVA: Hermann Buberl, Andrea Hölscher, Martin Maier, Melanie Strecker, Philine Raubold, 475 Lisa Rubin for support during sampling at CON and lab analysis; Stephan Raspe (LWF Bayern) for rainfall samples and meteorological data of MIT. Thanks also to our local partners at the three sites: Hans Amman, Siegfried Fink, Harry Peintinger, Paul Steinhard, Fabian Reichel, Hubert Waldmann, Othmar Winterhalder.

We also like to thank the Editor and three anonymous reviewers for their useful comments.

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

| Properties | MIT | CON | TUT |
|---|---|---|---|
| Soil | Hyperdystric chromic folic cambisol[a] | Hyperdystric skeletic folic cambisol[a] | Rendzic Leptosols [a] |
| Humus form (thickness in cm) | Moder (8) [b] | Mor-like Moder (13) [b] | Mull (12) |
| Soil texture (topsoil/subsoil) [a] | Loam / Sandy Loam [b] | Loam / Sandy Loam [b] | Clay / Clay |
| Stone content (topsoil/subsoil) [%] | 23/26 | 87/62 | 50/67 |
| pH ($H_2O$) (forest floor/ topsoil /subsoil) | 3.53 / 3.57 / 4.61 [b] | 3.46 / 4.03 / 4.61 [b] | 6.00 / 7.23 / 7.84 |
| Ptot stocks forest floor [g/m$^2$] | 7 [b] | 13 [b] | 19 |
| Ptot stocks mineral soil [g/m$^2$] | 624 [b] | 230 [b] | 189 |
| isotopically exchangeable P within 1 min (Ah, BA) [mg P/kg] | 9.2/3.0 [b] | 4.0/5.0 [b] | - / - |
| Parent material | Paragneiss [b] | Paragneiss [b] | Carbonate Rock |
| Slope [°] | 14° | 28° | 23° |
| Elevation [m a.s.l.] | 1023 [b] | 840 [b] | 835 |
| Dominant vegetation (mean age) | Fagus sylvatica (131 [b]) | Fagus sylvatica (132 [b]) | Fagus sylvatica (90 [c]) |
| Annual precipitation [mm/year] | 1299 [b] | 1749 [b] | 900 [c] |

**Tab. 2: Depth [cm] of lysimeter (LY) and trench (TR) installations aligned with the main soil horizons for Mitterfels (MIT), Conventwald (CON) and Tuttlingen (TUT); "B" indicates the sprinkling plot and is used to be consistent with other publications from the same sites. X indicates that LY4B and TR3B could not be installed at TUT due to shallow soil depth.**

| Depth [cm] | MIT | CON | TUT | Lysimeter, Trench |
|---|---|---|---|---|
| Forest floor | 0 | 0 | 0 | LY1B, TR1B |
| Topsoil | 35 | 40 | 40 | LY2B |
| Subsoil | 130 | 100 | 60 | LY3B, TR2B |
| Saprolite | 190 | 290 | X | LY4B, TR3B |

**Tab. 3: Mean pre-event water fraction [%] of the different flow components (trench: TR1B, TR2B, TR3B and lysimeter: LY1B, LY2B, LY3B, LY4B) and the 6 sprinkling experiments in Mitterfels (MIT), Conventwald (CON), Tuttlingen (TUT) in spring and summer. X indicates that this lysimeter or trench does not exist (TUT), NA indicates that a lysimeter or trench has not yielded any flow (MIT).**

| Event | LY1B | LY2B | LY3B | LY4B | TR1B | TR2B | TR3B |
|---|---|---|---|---|---|---|---|
| MIT Spring | 12 | 83 | 95 | 99 | 9 | 44 | NA |
| MIT Summer | 15 | 88 | 93 | 98 | 33 | 33 | NA |
| CON Spring | 10 | 69 | 63 | 86 | 19 | 63 | 78 |
| CON Summer | 16 | 58 | 83 | 78 | 24 | 55 | 74 |
| TUT Spring | 4 | 64 | 95 | X | 14 | 70 | X |
| TUT Summer | 4 | 83 | 70 | X | 14 | 47 | X |

**Tab. 4: Time to 20% rise in SSF [min] for all flow components and all sprinkling experiments. "–" indicates that time to 20% rise in SSF could not be calculated due to missing data at the beginning of the event (MIT in spring); X indicate that this LY or TR was not existing (TUT), NA indicates that this lysimeter or trench yielded no flow (MIT). Numbers in bold indicate cases where a flow component at deeper soil depth responded earlier than a flow component at shallower soil depth.**

| Event | LY1B | LY2B | LY3B | LY4B | TR1B | TR2B | TR3B |
|---|---|---|---|---|---|---|---|
| MIT Spring | – | 95 | 290 | 515 | 10 | 80 | NA |
| MIT Summer | 30 | 210 | 535 | 765 | 20 | 55 | NA |
| CON Spring | 15 | 160 | **105** | 220 | 35 | 315 | **235** |
| CON Summer | 10 | 70 | 180 | 235 | 25 | 305 | **265** |
| TUT Spring | 70 | 85 | 140 | X | 70 | 145 | X |
| TUT Summer | 15 | 205 | **140** | X | 20 | 295 | X |

**Tab. 5:** Time to 20% event water fraction [min] for all flow components and all sprinkling experiments; "–" indicates that time to 20% event water fraction could not be calculated due to missing data at the beginning of the event (MIT in spring); X indicates that this LY or TR was not existing (TUT), NA indicates that this trench yielded no flow (MIT), "/" indicates that 20 % event water fraction was not reached during the event; numbers in bold indicate that the time to 20% event water fraction was shorter than the time to 20% rise in SSF (see Tab. 4).

| Event | LY1B | LY2B | LY3B | LY4B | TR1B | TR2B | TR3B |
|---|---|---|---|---|---|---|---|
| MIT Spring | – | 110 | / | / | 15 | 250 | NA |
| MIT Summer | 45 | 565 | / | / | 30 | 75 | NA |
| CON Spring | 25 | 295 | 325 | 515 | 45 | **80** | **65** |
| CON Summer | 15 | 115 | 285 | 390 | 35 | **70** | **80** |
| TUT Spring | 75 | 300 | / | X | 80 | 380 | X |
| TUT Summer | 15 | 480 | 370 | X | 20 | **235** | X |

**Tab. 6:** Slope of the linear regression between Ptot concentration and fraction of event water for the lysimeters (LY) and the trenches (TR) and for all sprinkling experiments in Mitterfels (MIT), Conventwald (CON) and Tuttlingen (TUT): bold values are not significantly different from a slope = zero and therefore the relation likely chemostatic (Alpha = 0.05). Italic values indicate slopes that are not chemostatic and not significantly different from slopes describing simple dilution. Therefore, these relations are likely governed by proportional mixing. X indicate that this LY or TR was not existing (TUT), NA indicates that the trench yielded no flow (MIT).

| Event | LY1B | LY2B | LY3B | LY4B | TR1B | TR2B | TR3B |
|---|---|---|---|---|---|---|---|
| MIT Spring | -0.6 | 0.1 | **0.0** | 1.8 | **-0.2** | **0.0** | NA |
| MIT Summer | *-2.3* | **-0.3** | **-0.6** | **-1.5** | -0.1 | *-0.5* | NA |
| CON Spring | -2.2 | -1.2 | **0.0** | **0.0** | *-1.4* | **0.1** | **0.1** |
| CON Summer | **-0.1** | **0.0** | *-0.1* | **0.0** | -1.0 | **0.1** | **0.1** |
| TUT Spring | -0.1 | **0.0** | **0.0** | X | -0.1 | *0.0* | X |
| TUT Summer | *-0.5* | 0.1 | **0.0** | X | -0.6 | *-0.1* | X |

**Tab. 7:** Estimation of annual Ptot fluxes in vertical (LY) and lateral (TR) SSF at CON calculated based on the mean P concentration after the first flush during the sprinkling experiment at CON in summer (see Fig. 5) and the water flux between 10.05.2018 and 10.05.2019. This period has been chosen in order to not include the sprinkling experiments. No long-term measurements are available for MIT and TUT.

| | LY1B | LY2B | LY3B | LY4B | TR1B | TR2B | TR3B |
|---|---|---|---|---|---|---|---|
| Water flux [mm/a] | 295 | 239 | 320 | 102 | 2 | 4 | 1 |
| Median P concentration after first flush [mg/l] | 0.17 | 0.07 | 0.02 | 0.03 | 0.15 | 0.05 | 0.04 |
| Estimated P flux [kg/ha/a] | 0.49 | 0.16 | 0.07 | 0.03 | 0.00 | 0.00 | 0.00 |

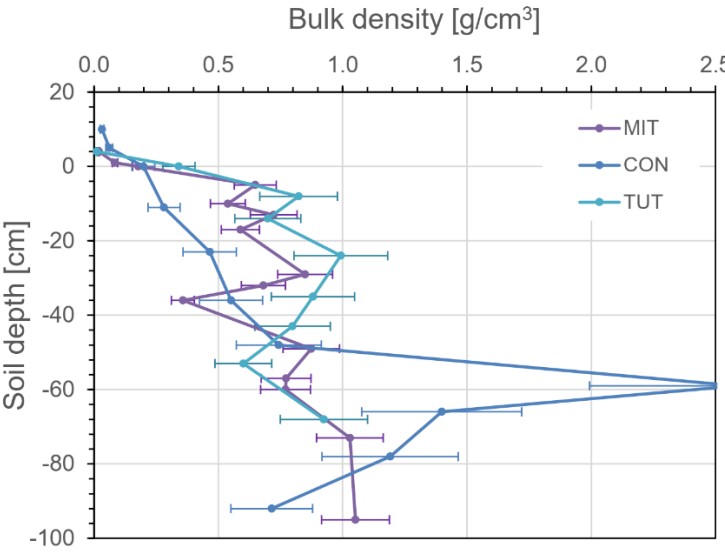

**Fig. 1: Depth profile of soil bulk density (mass fine earth per volume fine earth) of the three experimental sites Mitterfels (MIT), Conventwald (CON) and Tuttlingen (TUT). Due to the high stone contents of the soils the ''quantitative soil pit'' (QP) as described in** (Lang et al., 2017)**was used to determine bulk densities. Error bars represent standard deviation calculated based on the average coefficient of variance of the method determined at the corresponding study sites.**

**Fig. 2: Experimental setup to measure vertical and lateral subsurface flow with drainage pipes at three soil depths in a trench (TR1B, TR2B, TR3B) and zero tension lysimeters (i.e. steel piling plates pushed in the undisturbed soil profile from the side of the hillslope; LY1B, LY2B, LY3B, LY4B). "B" indicates the sprinkling plot and is used to be consistent with other publications from the same sites.**

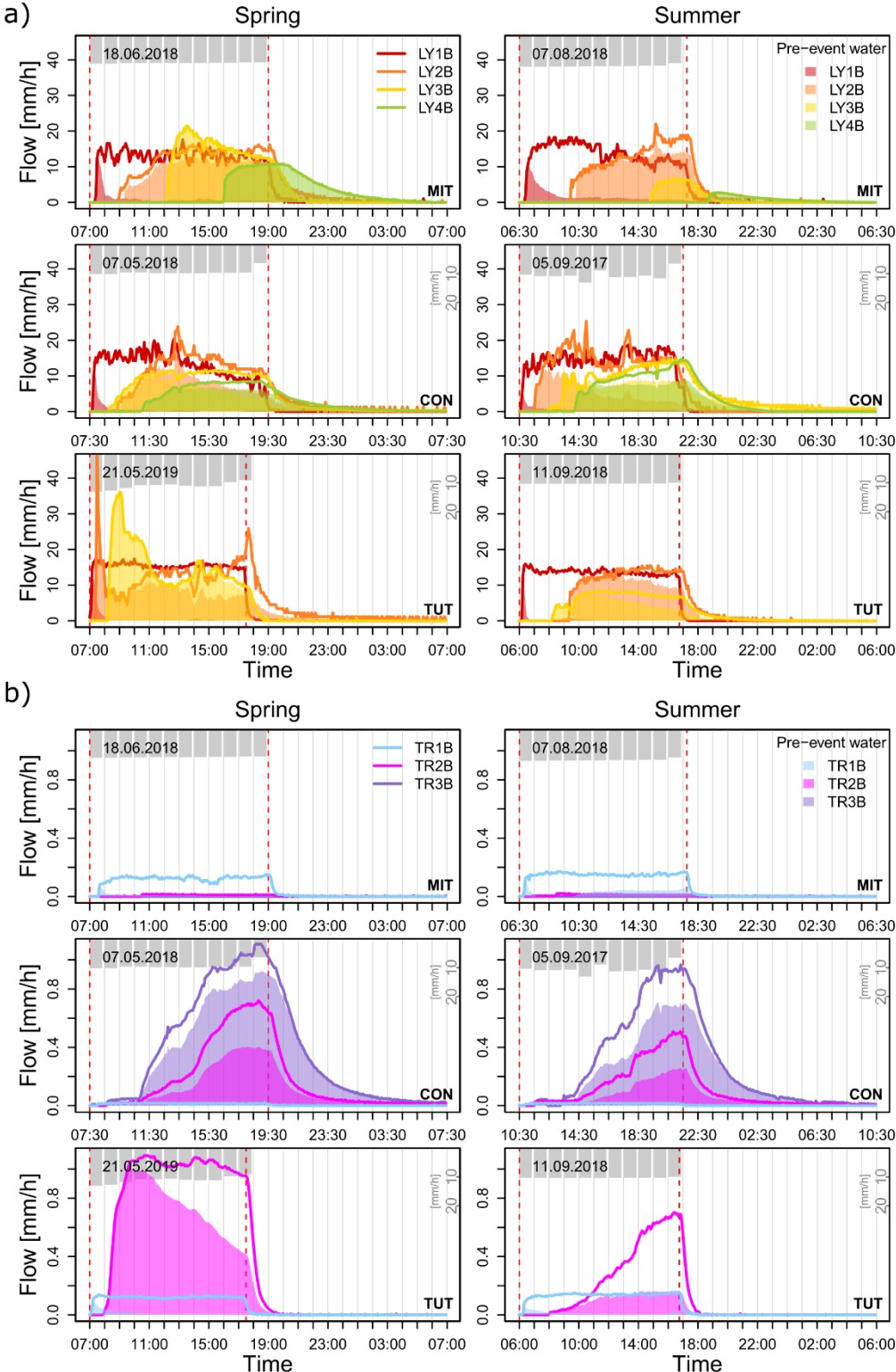

**Fig. 3: a) Vertical subsurface flow from the lysimeters (LY1B, LY2B, LY3B, LY4B) and b) lateral subsurface flow from the trench (TR1B, TR2B, TR3B) for the sprinkling experiments in Mitterfels (MIT), Conventwald (CON) and Tuttlingen (TUT) sorted in rows and for spring (left column) and summer (right column). Color-shaded areas show the pre-event water contribution to total discharge of each component (calculated from the Deuterium tracer). Red dashed lines indicate the start and end of the sprinkling experiment. Gray bars show sprinkling input during the experiments.**

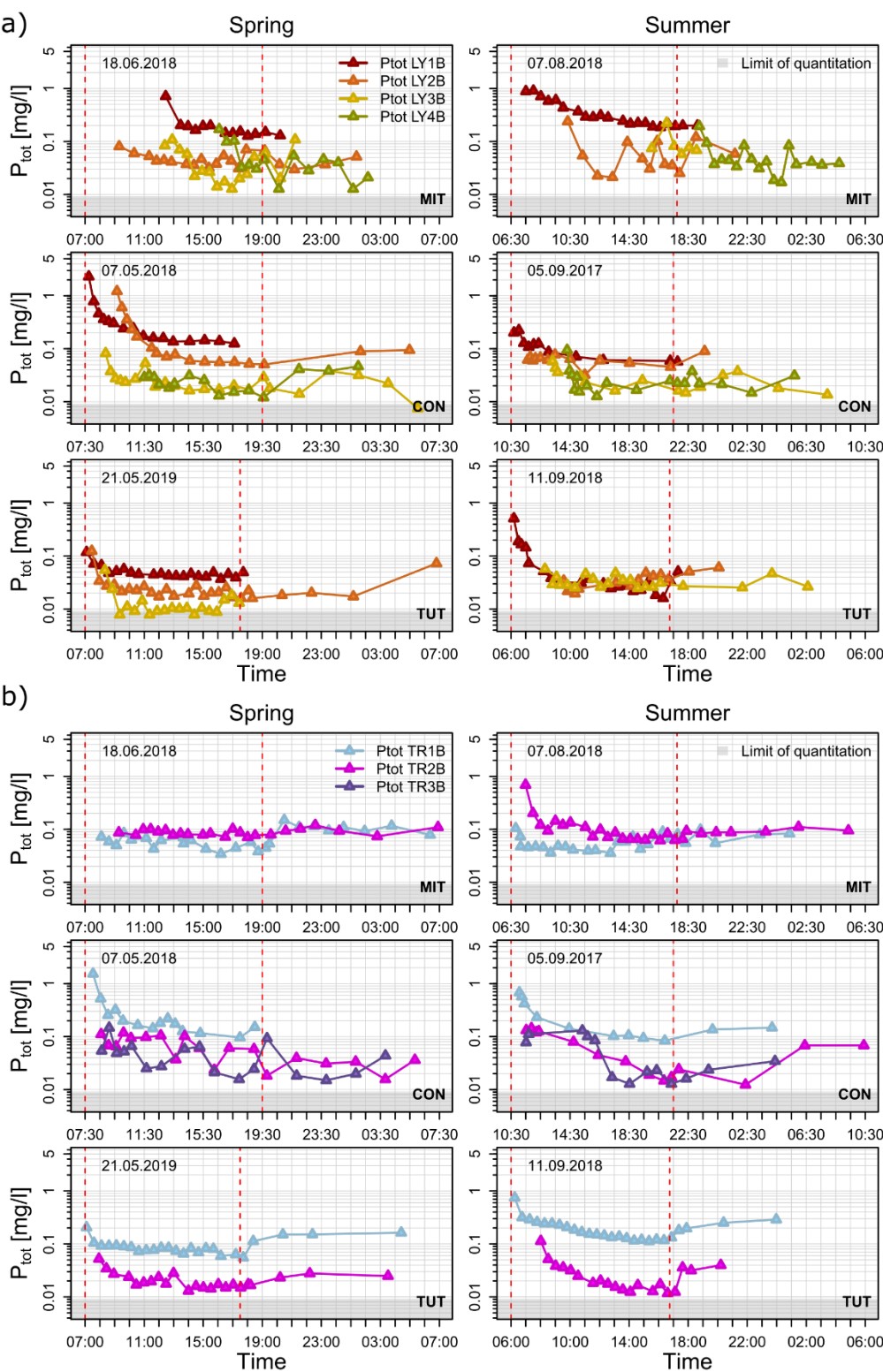

**Fig. 4: Ptot concentrations of (a) vertical subsurface flow and (b) lateral subsurface flow for the sprinkling experiments in Mitterfels (MIT), Conventwald (CON) and Tuttlingen (TUT) sorted in rows and for spring (left column) and summer (right column). Red dashed lines indicate the start and end of the sprinkling experiment, light gray areas indicate range of Ptot below limit of quantitation.**

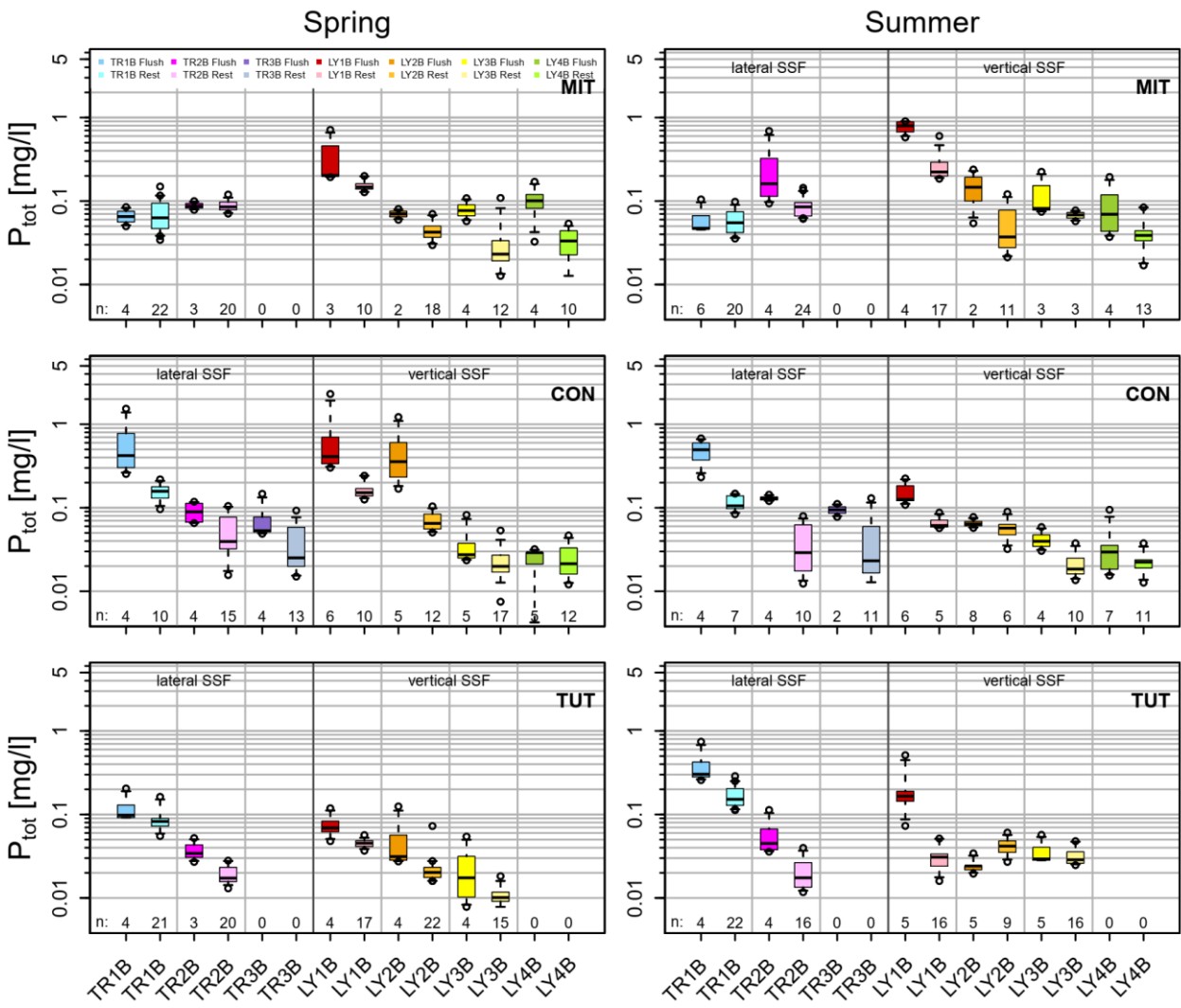

**Fig. 5: Ptot concentrations of vertical and lateral subsurface flow (SSF) during the flushing period defined as the first 2 h after onset of each flow component (darker colors) and the remaining part of the event (10 to 12 h) in lighter colors for the sprinkling experiments in Mitterfels (MIT), Conventwald (CON) and Tuttlingen (TUT).**

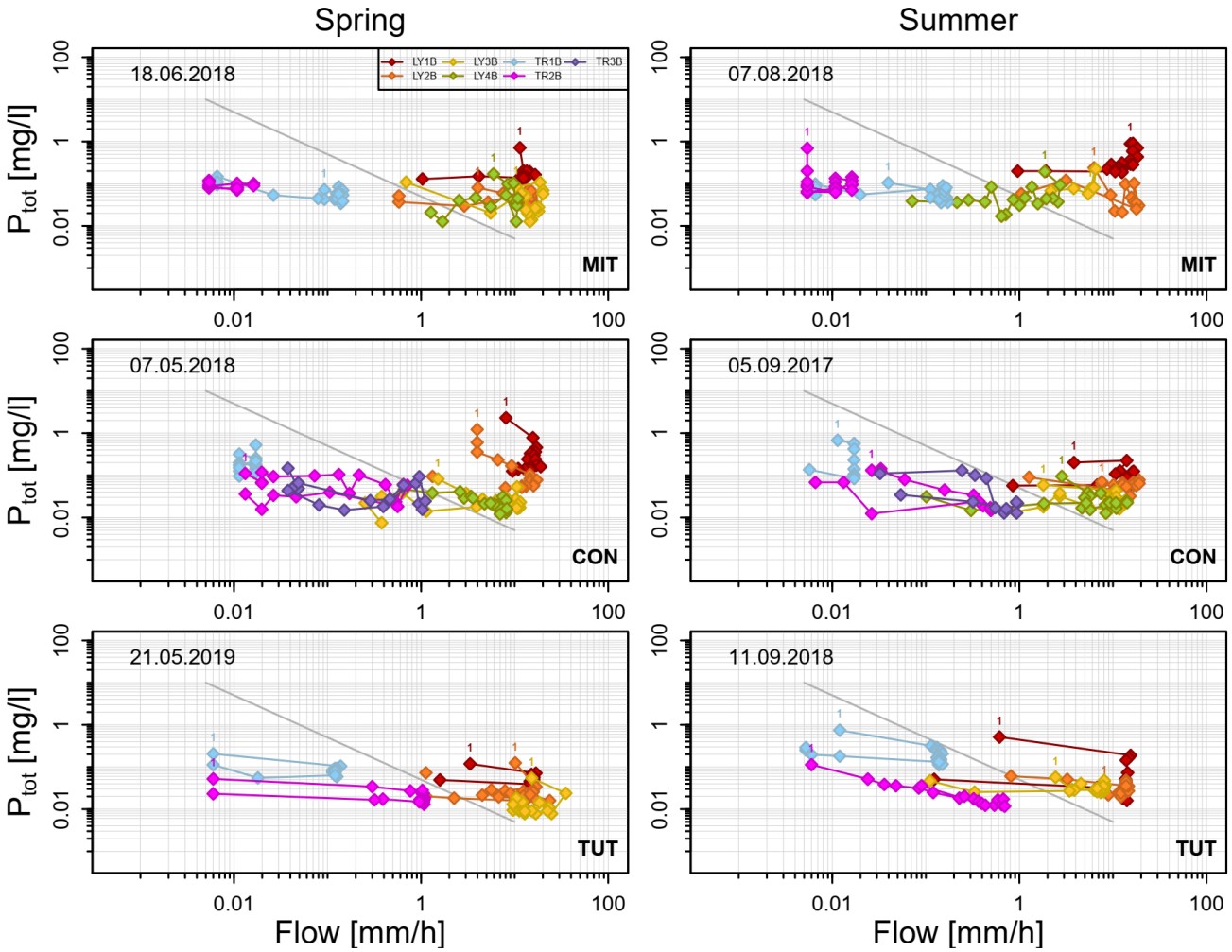

**Fig. 6: Ptot concentrations as a function of subsurface flow in the lysimeters (LY) and the trench (TR) during the sprinkling experiments in Mitterfels (MIT), Conventwald (CON) and Tuttlingen (TUT). The gray diagonal line indicates simple dilution (C ~ 1/Q). Label "1" indicates the first data point of each flow component.**

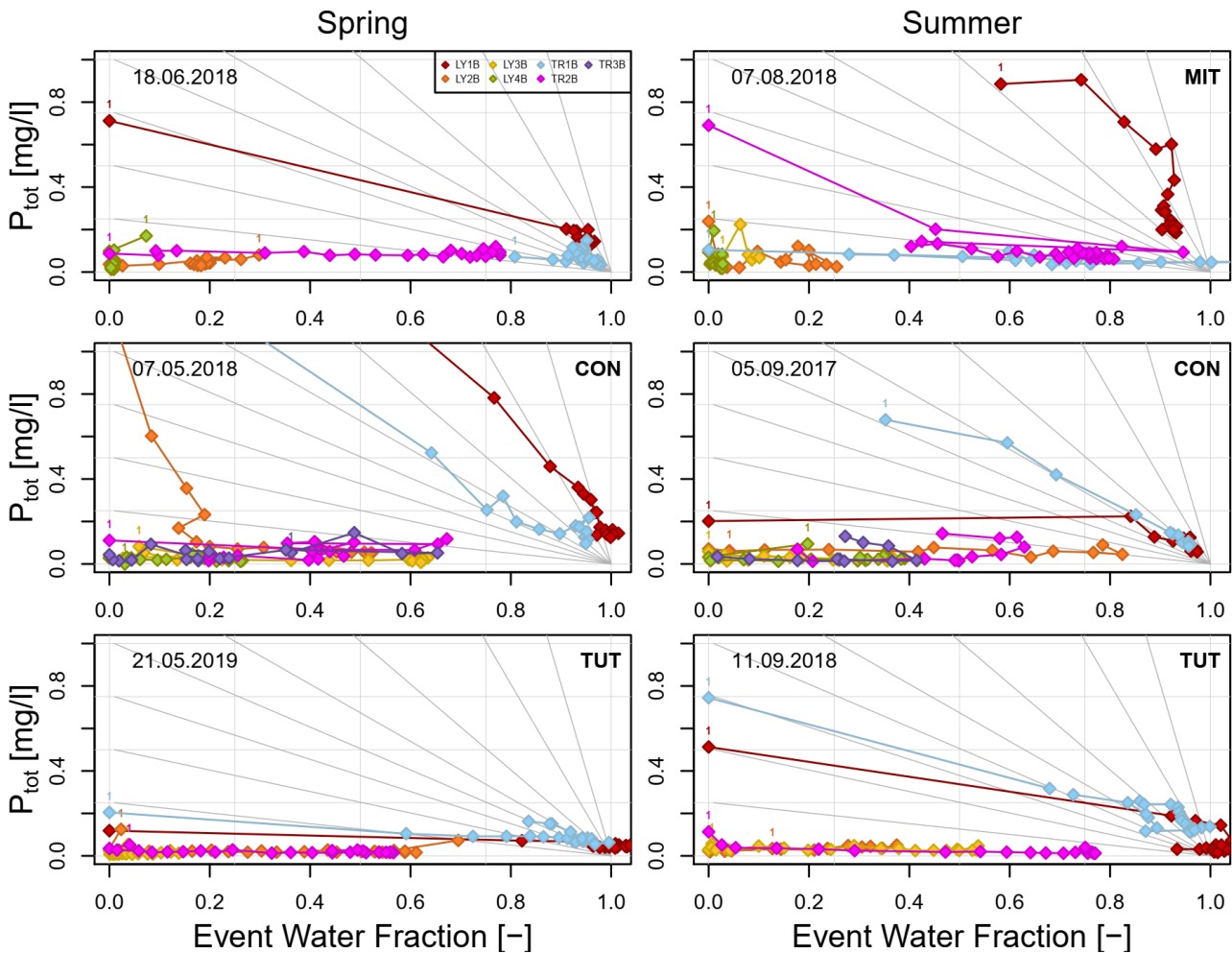

Fig. 7: Ptot concentrations as a function of event water fraction calculated for the sprinkling experiments. Gray lines indicate a selection of possible lines that describe the theoretical change in Ptot concentration assuming simple dilution (i.e., proportional mixing of event and pre-event water according to the event water fraction). Label "1" indicates the first data point of each flow component.

740

745