# Peer review of "Subsurface flow and phosphorus dynamics in beech forest hillslopes during sprinkling experiments: How fast is phosphorus replenished?"

_Biogeosciences, 2020_

## Referee Comment (RC1) · Anonymous Referee #2 · 11 May 2020

General comments The manuscript entitled " Phosphorus Transport in Subsurface Flow at Beech Forest Stands: Does Phosphorus Mobilization Keep up with Transport? ", written by Michael Rinderer, Jaane Krüger, Friederike Lang, Heike Puhlmann, and Markus Weiler, presents valuable results that contribute to the understanding of phosphorus transport in and phosphorus losses from the soil. The topic falls into the scope of Biogeosciences. The manuscript comprises results from large sprinkling experiments at three beech forest sites in Germany. The methods are adequate to test the research questions. The results are described in detail and can be used to answer the research question. The text is easily understandable, tables and figures are well-arranged and the conclusions are sound. Hence, I would recommend to consider this

manuscript for publication in Biogeosciences after minor revision.

Specific comments L 14 The values differ from those in Tab. 1. L75/76 The time oft he two experiments was not well chosen if microbial conditons – like soil moisture, temperature, litter fall – should differ. Rather late autumn/early winter (november; wet, cold, a lot of litter) and summer (july/august; dry, warm, less litter) should have been chosen. L 227 trise20 of the event water fraction is in Tab. 5 and trise20 of SSF in Tab. 4 L 233-252 Results oft he statistical analyses are not displayed anywhere and the statistical approach is not described in the materials and methods section. L 295/296 "A peak of high event water at the beginning oft he sprinkling experiments, . . ." I could not find this result in the presented data (Fig. 3?). L 302 Tab. 1 (skeleton content) and Fig. 1 (soil bulk density) L 315-317 Why is the Ptot concentration from the mineral soil in vertical SSF in MIT lower than from the forest floor? L 339/340 This is only true for vertical SSF, isn't it? L345 This is predominately the case for LY1B, isn't it (Suppl. Tab. 1)? L 350/351 Which soil properties? L 361 It is unlikely that adsorption explains the difference, since adsorption is very small in the forest floor. How large was the P flow from the 3 sites in g/m2 (in cormparison tot he soil P stocks oft he 3 sites)? Compare it with values from the literature that you cited in the introduction (L 30 and others). Tab. 2 Why was the soil depth oft he installations in the subsoil in CON different from MIT? Tab. 4 and 5 You abbreviate both varaiables with trise20; better add "in SSF" in Tab. 4 and "in event water fraction" in Tab. 5. Fig. 3b Reorder the labels (TR1B, TR2B, TR3B) according to the labels in Fig. 3a (from forest floor to saprolite). Fig. 5 The unit of the flow on the x-axis is mm/h, isn' it?

Technical corrections L 29 forest ecosystem -> forest ecosystems L 66 In biopores -> Biopores L 171 chemotatic -> chemostatic L 225 paranthesis is missing L 287 suction caps -> suction cups L 332 Makoswski et al -> Makowski et al. L 338 suggest -> suggests L 345 was -> were L 357 and 370 expect -> except Tab. 1 Dominant vegetation and Annual precipitation for TUT: d -> c Fig. 5 and Fig. 6 Labels -> Label

Please also note the supplement to this comment:
https://www.biogeosciences-discuss.net/bg-2020-118/bg-2020-118-RC1-supplement.pdf

---

## Referee Comment (RC2) · Anonymous Referee #1 · 15 May 2020

General remark

The authors present data from sprinkling experiment in three forest sites, performed during two different seasons, where they analyzed water flow and soil solution P concentrations. The paper is generally well-written and easy to follow, and the results are interesting. However, from my point of view the motivation/objective of the paper is not yet properly addressed with the results. This needs to be addressed before the manuscript can be published.

The stated objective of this paper to quantify P losses via subsurface flow (abstract, as well as l. 75 of introduction). This sets the reader up to expect to learn about

phosphorus fluxes. More information on subsurface flow P losses would indeed be very interesting, also for the land surface modelling community, which is struggling to incorporate P cycling into C, N models. However, in the results no soil P fluxes [g P m-2 time-1] are presented, only P concentrations [mg P L-1 water]. I suggest authors to bring the paper in line with the objectives. Firstly, in the introduction by introducing what are typical soil P stocks in forest ecosystems (see e.g. (Achat et al. 2016; Hou et al. 2018)), and further what are orders of magnitude for P flux losses (e.g. in g P m-2 yr-1) as determined by earlier studies (see e.g. (Vitousek 2004) and others—authors would have to search the literature a bit here). Perhaps also comparing to other P fluxes in forest ecosystems such as dust deposition, rock weathering, etc. This will set the scene for talking about P fluxes in forest ecosystems. I'm guessing that the losses will be several orders of magnitude lower than the stocks, and it will have to be argued why (if?) they are still important.

Secondly, no P flux data is presented in the results. Is it possible to multiply water flow by P conc. to get P flux? Why is this not done?

The discussion should be developed further also. How do the results from this study tie into what we already know about P cycling in forests, and P loss pathways? At the moment the discussion mostly explains the results, but it needs to go further to show readers what has been learned. Again, given the setup of the paper, the focus should be on P fluxes. What do the results mean in terms of fluxes? What do we learn about P cycling in forest ecosystems? Just thinking out loud (authors may choose to follow up on this or not): Apart from the nutrient flush in the first 1-2 hours, P concentrations were relatively constant regardless of SSF. On a methodological note, does this imply that we can (roughly) approximate annual P losses via SSF given the water balance of the site and the soil solution P concentration? What would that imply in terms of annual P loss [g P m-2 yr-1] for these sites? How does that compare to the forest stocks and orders of magnitude that can be expected for other loss and input pathways such as dust deposition, weathering and erosion (Chadwick et al. 1999; Hartmann et al. 2014;

Tipping et al. 2014; Aciego et al. 2017)?

Specific comments

Title: This is up to the authors, but if they want their article to also reach hydrologists, the title (and abstract?) should be revised. A good portion of the results and discussion as well as the conclusion focus on water flow, which I did not expect from reading the title. E.g. something along the lines of "Beech forest stands sprinkling experiments: effects on sub-surface flow and phosphorus dynamics"

l. 23 Jumping on the "climate change" bandwagon here is unwarranted. There is no discussion of climate change in the article. Also, the data rather show that P conc. is constant and thus only dependent on water balance, right?

l. 29 How much P is in forest soils? How big are these losses?

l. 32 remove period after "SSF"

l. 34 remove period after "nutrients"

l. 45 The way this sentence is written makes it sound like it was done in this study. I suggest to change tense to "has been" or state "in previous studies"

l. 52 add ", USA"

l. 54-62 This is too detailed and should be condensed

l. 66 "In biopores..." ?

l. 74 "We performed. . ..to capture potential differences in P fluxes." However, in the research questions the focus is on dynamics of P concentrations. This should be aligned.

l. 99 231 g at CON is very similar to 209 g at TUT, especially given heterogeneity inherent to soils. I don't think you can argue that TUT is "less rich in soil P" than CON.

l. 99 So that the reader can put these numbers into relation (is 209 – 678 g P m2 really a large range in P, justifying calling one P poor and the other P rich?), I again suggest

presenting orders of magnitude ranges in soil P stocks in forests (see comment l. 29)

l. 102 Add period before "Bulk". I stop correcting spelling / grammar mistakes at this point, but there are more in the remaining text. Please proof read the next version carefully.

l. 136 I'm no expert here, but I'm guessing rain water is far from de-ionized. How do you think using deionized water affected the results? Does that need to be discussed?

l. 170-177 Nice setup to let the reader now what to expect, look for and interpret in the results! That's an example of great scientific writing

Table 1: Please also add pH to the table. pH is an important indicator of soil P forms and dynamics and may be important to explain the results, e.g. the difference between TUT, CON and MIT.

Fig. 1 and others: colors are not grayscale print-friendly

Fig. 2: Very nice overview figure. This makes it a lot easier to understand what was done.

Fig 3. also this is a nice figure. I suggest to move spring before summer. I understand that spring experiments were carried out a year later, and that's ok since you have the dates there and it can be noted in the figure caption. But it makes more sense to have the plots in seasonal order for interpreting the plots

Section 3.4 It would have been interesting to measure inorganic and organic P as opposed to only total P.

Results section 3.5: multiypling conc. by water flow = element flux. Why not present these data in a section 3.5 "Soil P fluxes"

l. 256-260 (p. 8-9): I'm not surprised that P conc. in the soil solution remains relatively constant. If we consider the very fast turnover time of P in the soil solution of only seconds to minutes (Helfenstein et al. 2018).

l. 300 What about biopores? Is there evidence to suggest that CON and TUT have more earthworms or other large soil fauna

section 4.2 This section could be re-written to make it more focused. At the moment there is a mix of rather trivial findings, such as that P stocks are higher in the forest floor than in the mineral soil, while the interesting things are not discussed in-depth enough. The discussion of P concentration dynamics should be better linked to existing literature, e.g. what is known about turnover time of P in the soil solution and phosphate buffering capacity. Phosphorus-buffering capacity (PBC) is defined as the ability of soil to moderate changes in the concentration of soil solution P (Beckett and White 1964; Olsen and Khasawneh 1980; Barrow 1983; Pypers et al. 2006), and would be interesting to bring in here. Soil solution P turnover, a related concept, has been shown to be negatively correlated with P conc. in the soil solution (Helfenstein et al. 2018), which authors might consider discussing as well. (i.e. the more P in the soil solution (forest floor), the slower the turnover time; the less P in the soil solution (mineral soil), the faster the turnover time.

l. 347 As with the plots, I would take spring before summer.

l. 364 not exactly true that you have six different experiments. It's one experiment carried out on three sites and at two time points.

l. 371 It's quite well known that soil solution P concentrations are lower with increasing soil depth. I would rather focus on novel findings in the conclusion.

l. 372 "it was especially strong. . ." What is it?

l. 373 It is obvious that P concentrations are highest in the P-rich site. Again, the conclusion should focus on the novel findings.

l. 374 "Particulalry high". Please be concrete. How much higher? Are we talking 1.5x, 2x or 10x higher than during the rest of the experiment?

l. 375 – 379 This is interesting and in my opinion the main finding of the study. This

should be placed more prominently and discussed appropriately.

l. 380 Conclusion not supported by the data. There was no discussion of climate change in the article.

l. 436 "DWD, 2010" please provide complete citation reference

References Achat DL, Pousse N, Nicolas M, et al (2016) Soil properties controlling inorganic phosphorus availability: general results from a national forest network and a global compilation of the literature. Biogeochemistry 127:255–272. doi: 10.1007/s10533-015-0178-0 Aciego SM, Riebe CS, Hart SC, et al (2017) Dust outpaces bedrock in nutrient supply to montane forest ecosystems. Nat Commun 8:14800. doi: 10.1038/ncomms14800 Barrow NJ (1983) A mechanistic model for describing the sorption and desorption of phosphate by soil. J Soil Sci 34:733–750. doi: 10.1111/j.1365-2389.1983.tb01068.x Beckett PHT, White RE (1964) Studies on the phosphate potentials of soils. Plant Soil 21:253–282. doi: 10.1007/bf01377744 Chadwick OA, Derry LA, Vitousek PM, et al (1999) Changing sources of nutrients during four million years of ecosystem development. Nature 397:491–497. doi: 10.1038/17276 Hartmann J, Moosdorf N, Lauerwald R, et al (2014) Global chemical weathering and associated P-release — The role of lithology, temperature and soil properties. Chem Geol 363:145–163. doi: 10.1016/j.chemgeo.2013.10.025 Helfenstein J, Jegminat J, McLaren TI, Frossard E (2018) Soil solution phosphorus turnover: derivation, interpretation, and insights from a global compilation of isotope exchange kinetic studies. Biogeosciences 15:105–114. doi: 10.5194/bg-15-105-2018 Hou E, Tan X, Heenan M, Wen D (2018) A global dataset of plant available and unavailable phosphorus in natural soils derived by Hedley method. Sci Data 5:180166. doi: 10.1038/sdata.2018.166 Olsen SR, Khasawneh FE (1980) Use and limitations of physical-chemical criteria for assessing the status of phosphorus in soils. In: Khasawneh FE, Sample EC, Kamprath EJ (eds) The Role of Phosphorus in Agriculture. American Society of Agronomy, Crop Science Society of America, Soil Science Society of America, Madison, WI, pp 361–410 Pypers P, Delrue J, Diels J, et al (2006) Phosphorus intensity determines

short-term P uptake by pigeon pea (Cajanus cajan L.) grown in soils with differing P buffering capacity. Plant Soil 284:217–227. doi: 10.1007/s11104-006-0051-y Tipping E, Benham S, Boyle JF, et al (2014) Atmospheric deposition of phosphorus to land and freshwater. Environ Sci Process Impacts 16:1608–1617. doi: 10.1039/c3em00641g Vitousek PM (2004) Nutrient Cycling and Limitation: Hawai'i as a Model System. Princeton University Press, Princeton

―――――――――――――――――

---

## Author Comment (AC1) · 22 Jun 2020

We thank the reviewer1 for his/her assessment of our paper manuscript and the useful comments to improve the text. We have uploaded a pdf file as supplement that provides a response (in red color) to each of the comments suggested by the reviewer. This way we hope to guarantee a better readability.

Please also note the supplement to this comment:
https://www.biogeosciences-discuss.net/bg-2020-118/bg-2020-118-AC1-supplement.pdf

**Supplement:**

*We thank the reviewer1 for his/her assessment of our paper manuscript and the useful comments to improve the text. We have uploaded a pdf file as supplement that provides a response (in red color) to each of the comments suggested by the reviewer. This way we hope to guarantee a better readability.*

General remark: The authors present data from sprinkling experiment in three forest sites, performed during two different seasons, where they analyzed water flow and soil solution P con-centrations. The paper is generally well-written and easy to follow, and the results are interesting. However,  from my point of view the motivation/objective of the paper is not yet properly addressed with the results.  This needs to be addressed before the manuscript can be published. The stated objective of this paper to quantify P losses via subsurface flow (abstract, as  well  as  l.  75  of introduction). This  sets  the  reader  up  to  expect  to  learn  about phosphorus fluxes.  More information on subsurface flow P losses would indeed be  very interesting, also for the land surface modelling community, which is struggling to incorporate P cycling into C, N models.  However, in the results no soil P fluxes [g Pm-2 time-1] are presented, only P concentrations [mg P L-1 water].  I suggest authors to bring the paper in line with the objectives. Firstly, in the introduction by introducing what are typical soil P stocks in forest ecosystems (see e.g. (Achat et al. 2016; Hou etal. 2018)), and further what are orders of magnitude for P flux losses (e.g. in g P m-2yr-1) as determined by earlier studies (see e.g. (Vitousek 2004) and others authors would have to search the literature a bit here).   Perhaps also comparing to other Pfluxes in forest ecosystems such as dust deposition, rock weathering, etc. This will set the scene for talking about P fluxes in forest ecosystems. I'm guessing that the losses will be several orders of magnitude lower than the stocks, and it will have to be argued why (if?) they are still important. Secondly, no P flux data is presented in the results. Is it possible to multiply water flowby P conc. to get P flux? Why is this not done?

We agree that P-fluxes are an interesting theme. In fact, we are preparing another paper manuscript that is addressing this topic. In the paper we submitted here we want to describe the experimental setup and focus on P-concentrations, the nutrient flushing and the chemostatic behavior towards the end of the experiments. We argue that there is not much literature that presents soil-depth specific P concentrations in high temporal resolution from forest stands measured under field conditions why we think it is worth to present this data and have a separate paper on the fluxes. As a response to the reviewer we however will consider to reformulate the title to be clearer.

The discussion should be developed further also. How do the results from this study tie into what we already know about P cycling in forests, and P loss pathways? At the moment the discussion mostly explains the results, but it needs to go further to show readers what has been learned. Again, given the setup of the paper, the focus should be on P fluxes. What do the results mean in terms of fluxes? What do we learn about P cycling in forest ecosystems?

We will revisit the discussion section and better address the topics suggested. P fluxes however are the topic of a second paper currently in preparation.

Just thinking out loud (authors may choose to followup on this or not): Apart from the nutrient flush in the first 1-2 hours, P concentrations were relatively constant regardless of SSF. On a methodological note, does this imply that we can (roughly) approximate annual P losses via SSF given the water balance of the site and the soil solution P concentration? What would that imply in terms of annual P loss [g P m-2 yr-1] for these sites? How does that compare to the forest stocks and orders of magnitude that can be expected for other loss and input pathways such as dust deposition, weathering and erosion (Chadwick et al. 1999; Hartmann et al. 2014; Tipping et al. 2014; Aciego et al. 2017) ?

We are developing similar thought and will support these with data analysis but this is subject of a different paper manuscript in preparation.

Specific comments

Title: This is up to the authors, but if they want their article to also reach hydrologists, the title (and abstract?) should be revised. A good portion of the results and discussion as well as the conclusion

focus on water flow, which I did not expect from reading the title.  E.g.  something along the lines o f "Beech forest stands sprinkling experiments: effects on sub-surface flow and phosphorus dynamics"

We will consider changing or adapting the title to address a broader audience. A revision will also update the abstract.

l.  23 Jumping on the "climate change" bandwagon here is unwarranted.  There is no discussion of climate change in the article.  Also, the data rather show that P conc. is constant and thus only dependent on water balance, right?

The idea of this sentence was to put the paper in a very broad, general context but we agree that climate change is not a main theme in the following paper analysis. Still, precipitation is predicted to change as a consequence of climate change, and by this will have an effect on SSF and thus P-transport. We will edit this part to be clearer.

l. 29 How much P is in forest soils? How big are these losses?

We will address P-stocks and fluxes in the second paper.

l. 32 remove period after "SSF"

We will address that

l. 34 remove period after "nutrients"

We will address that

l.  45 The way this sentence is written makes it sound like it was done in this study.  I suggest to change tense to "has been" or state "in previous studies"

We will address that

l. 52 add ", USA"

We will address that

l. 54-62 This is too detailed and should be condensed

We will address that

l. 66 "In biopores…" ?

We will address that

l. 74 "We performed….to capture potential differences in P fluxes." However, in the research questions the focus is on dynamics of P concentrations. This should be aligned.

We will address that

l. 99 231 g at CON is very similar to 209 g at TUT, especially given heterogeneity inherent to soils. I don't think you can argue that TUT is "less rich in soil P" than CON.

We realize that the way we wrote the sentence is maybe misleading to the reader. We will state the P-content but not rank it relative to each other.

l. 99 So that the reader can put these numbers into relation (is 209 – 678 g P m2 really a large range in P, justifying calling one P poor and the other P rich?), I again suggest presenting orders of magnitude ranges in soil P stocks in forests (see comment l. 29)

We argue that between the MIT (ca. 700 g P m2) and the other two sites ( ca. 200 g P m2) there is a factor of >1/3. At least for forests in Central Europe this is a significant difference. However, we agree that we can avoid the terms "rich" and "poor" and talk about "higher" and "lower" instead.

l. 102 Add period before "Bulk". I stop correcting spelling / grammar mistakes at this point, but there are more in the remaining text. Please proof read the next version carefully.

We will work on this

l. 136 I'm no expert here, but I'm guessing rain water is far from de-ionized. How do you think using deionized water affected the results? Does that need to be discussed?

Collecting 60.00 L of rainfall for the experiment was not an option. So, we were left with using groundwater from the drinking supply system. We argue that using untreated groundwater as sprinkling water would have been unacceptable from an experimental design point of view simply because it is an unnatural source of hydrochemical compounds (including P) to the system. We think that the term "deionized" might make some readers think of purified water like in a lab environment. To show that this was not the case, we had added in L 136 that the water had an electrical conductivity of 20 µS/cm. This is comparable to some natural rainfall. The 20 µS/cm is a

result of the efficiency of the industrial deionizer and processing 60.000 L of water. However, to avoid irritation, we will avoid the term "deionized water" as much as possible.

l. 170-177 Nice setup to let the reader now what to expect, look for and interpret in the results! That's an example of great scientific writing .

We appreciate your positive comment

Table 1: Please also add pH to the table. pH is an important indicator of soil P forms and dynamics and may be important to explain the results, e.g. the difference between TUT, CON and MIT.

We will add ph to Tab 1

Fig. 1 and others: colors are not grayscale print-friendly:

We argue to keep figures in color-scheme as showing all in gray scale is even harder to indicate the information included in the plots.

Fig. 2 Very nice overview figure. This makes it a lot easier to understand what was done.

We appreciate your positive comment

Fig 3. also this is a nice figure. I suggest to move spring before summer. I understand that spring experiments were carried out a year later, and that's ok since you have the dates there and it can be noted in the figure caption. But it makes more sense to have the plots in seasonal order for interpreting the plots

We consider changing the order of the final graphs.

Section 3.4 It would have been interesting to measure inorganic and organic P as opposed to only total P.

We see the reviewer's point but Ptot is the data that we have at hand.

Results section 3.5: multiypling conc. by water flow = element flux. Why not present these data in a section 3.5 "Soil P fluxes"

We prepare a second paper that has the focus on P fluxes.

l. 256-260 (p. 8-9): I'm not surprised that P conc. in the soil solution remains relatively constant. If we consider the very fast turnover time of P in the soil solution of only seconds to minutes (Helfenstein et al. 2018).

We will include this in the discussion

l. 300 What about biopores?  Is there evidence to suggest that CON and TUT have more earthworms or other large soil fauna

Biopores can also make a contribution to preferential flow. We will address this in the text. Due to the low pH, there are no or only a few earthworms to be found at MIT and CON. In TUT, earthworms are present. However, due to the clay content, we consider it very likely that cracks originating from shrinking and swelling processes make the largest contribution to preferential flow.

section 4.2 This section could be re-written to make it more focused.  At the moment there is a mix of rather trivial findings, such as that P stocks are higher in the forest floor than in the mineral soil, while the interesting things are not discussed in-depth enough. The discussion of P concentration dynamics should be better linked to existing literature, e.g. what is known about turnover time of P in the soil solution and phosphate buffering capacity. Phosphorus-buffering capacity (PBC) is defined as the ability of soil to moderate changes in the concentration of soil solution P (Beckett and White1964; Olsen and Khasawneh 1980; Barrow 1983; Pypers et al.  2006), and would be  interesting to bring in here. Soil solution P turnover, a related concept, has been shown to be negatively correlated with P conc.  in the soil solution (Helfenstein et al.  2018),which authors might consider discussing as well.  (i.e.  the more P in the soil solution(forest floor), the slower the turnover time; the less P in the soil solution (mineral soil),the faster the turnover time.

Thank you for your valuable input. We agree that including the mentioned issues can improve the discussion. We will revise this section

l. 347 As with the plots, I would take spring before summer.

We consider to change this in the final version

l.  364 not exactly true that you have six different experiments.   It's one experiment carried out on three sites and at two time points.

We will change this to be more precise

l. 371 It's quite well known that soil solution P concentrations are lower with increasing soil depth. I would rather focus on novel findings in the conclusion.

We will rewrite the text

l. 372 "it was especially strong..." What is it?l.  373 It is obvious that P concentrations are highest in the P-rich site.   Again,  the conclusion should focus on the novel findings.

We will rewrite the text and remove parts that are meant to summarize the paper.

l. 374 "Particulalry high". Please be concrete. How much higher? Are we talking 1.5x,2x or 10x higher than during the rest of the experiment?

We will change this to be more precise

l. 375 – 379 This is interesting and in my opinion the main finding of the study.  This should be placed more prominently and discussed appropriately.

We will extend this part

l. 380 Conclusion not supported by the data.  There was no discussion of climatechange in the article.

The last sentence of the paper will be removed.

l. 436 "DWD, 2010" please provide complete citation reference

Will be completed

References

Thanks for pointing out to these references. They are valuable for our paper

Achat DL, Pousse N, Nicolas M, et al (2016) Soil properties control-ling inorganic phosphorus availability:  general results from a national forest net-work and a global compilation of the literature.  Biogeochemistry 127:255–272.  doi:10.1007/s10533-015-0178-0

Aciego SM, Riebe CS, Hart SC, et al (2017) Dust out-paces bedrock in nutrient supply to montane forest ecosystems. Nat Commun 8:14800.doi: 10.1038/ncomms14800 Barrow NJ (1983) A mechanistic model for describingthe sorption and desorption of phosphate by soil.  J Soil Sci 34:733–750.   doi:10.1111/j.1365-2389.1983.tb01068.x

Beckett PHT, White RE (1964) Studies on thephosphate potentials of soils. Plant Soil 21:253–282. doi: 10.1007/bf01377744

Chad-wick OA, Derry LA, Vitousek PM, et al (1999) Changing sources of nutrients during fourmillion years of ecosystem development.  Nature 397:491–497.  doi: 10.1038/17276

Hartmann J, Moosdorf N, Lauerwald R, et al (2014) Global chemical weathering and associated P-release ã̌AˇT The role of lithology, temperature and soil properties. ChemGeol 363:145–163.  doi: 10.1016/j.chemgeo.2013.10.025

Helfenstein J, Jegminat J,McLaren TI, Frossard E (2018) Soil solution phosphorus turnover: derivation, inter-pretation, and insights from a global compilation of isotope exchange kinetic studies. Biogeosciences 15:105–114. doi: 10.5194/bg-15-105-2018

Hou E, Tan X, Heenan M,Wen D (2018) A global dataset of plant available and unavailable phosphorus in natu-ral soils derived by Hedley method. Sci Data 5:180166. doi: 10.1038/sdata.2018.166

Olsen SR, Khasawneh FE (1980) Use and limitations of physical-chemical criteria for assessing the status of phosphorus in soils. In: Khasawneh FE, Sample EC, Kam-prath EJ (eds) The Role of Phosphorus in Agriculture. American Society of Agronomy,Crop Science Society of America, Soil Science Society of America, Madison, WI, pp361–410

Pypers P, Delrue J, Diels J, et al (2006) Phosphorus intensity determines short-term P uptake by pigeon pea (Cajanus cajan L.) grown in soils with differing Pbuffering capacity. Plant Soil 284:217–227. doi: 10.1007/s11104-006-0051-y

Tipping E, Benham S, Boyle JF, et al (2014) Atmospheric deposition of phosphorus to land andfreshwater. Environ Sci Process Impacts 16:1608–1617. doi: 10.1039/c3em00641gVitousek PM (2004) Nutrient Cycling and Limitation: Hawai'i as a Model System. Princeton University Press, Princeton

---

## Author Comment (AC2) · 22 Jun 2020

We thank reviewer 2 for his/her positive assessment of our paper manuscript and the useful comments to improve the text. In the following we respond to each comment. Also here we have uploaded a pdf file as supplement that provides a response (in red color) to each of the comments suggested.

Please also note the supplement to this comment:
https://www.biogeosciences-discuss.net/bg-2020-118/bg-2020-118-AC2-supplement.pdf

[Figure]

[Figure]

**Supplement:**

**Reviewer 2**

**Anonymous Referee #2:**

General comments The manuscript entitled " Phosphorus Transport in Subsurface Flowat Beech Forest Stands: Does Phosphorus Mobilization Keep up with Transport? ",written by Michael Rinderer, Jaane Krüger, Friederike Lang, Heike Puhlmann, and Markus Weiler, presents valuable results that contribute to the understanding of phosphorus transport in and phosphorus losses from the soil. The topic falls into the scope of Biogeosciences. The manuscript comprises results from large sprinkling experiments at three beech forest sites in Germany. The methods are adequate to test the research questions. The results are described in detail and can be used to answer the research question. The text is easily understandable, tables and figures are well-arranged and the conclusions are sound. Hence, I would recommend to consider this manuscript for publication in Biogeosciences after minor revision.

We thank the reviewer2 for his/her positive assessment of our paper manuscript and the useful comments to improve the text. In the following we respond to each comment. Also here we have uploaded a pdf file as supplement that provides a response (in red color) to each of the comments suggested.

**Specific comments**

L 14 The values differ from those in Tab. 1.

**We corrected the values to match Table 1.1**

L75/76 The time of the two experiments was not well chosen if microbial conditons – like soil moisture, temperature, litter fall – should differ. Rather late autumn/early winter (november; wet, cold, a lot of litter) and summer (july/august; dry, warm, less litter) should have been chosen.

We agree that a stronger contrast in seasonality would have been better to evaluate seasonal effects. However, this was a sub-ordinate part of the study and is therefore not listed as a separate research question in our paper. Still we think it is interesting to mention our results in the text. When choosing our days of sprinkling we were restricted to the vegetation period (i.e., the time when trees had leaves and active photosynthesis) as we were also monitoring tree water uptake and P-transport in trees during the subsequent 4 to 6 weeks after the sprinkling experiment (papers in preparation).

However, we will rewrite the text as follows:

"We performed two sprinkling experiments at each site to capture potential differences in P fluxes within the vegetation period (i.e., between summer/fall and spring). ..."

And we deleted the part "...and litter fall is not evenly distributed across the year." from the manuscript.

L 227 trise20 of the event water fraction is in Tab. 5 and trise20 of SSF in Tab.4

Thanks for pointing this out. We corrected it.

L 233-252 Results of the statistical analyses are not displayed anywhere and the statistical approach is not described in the materials and methods section.

In addition to Figure 4 we will add another figure that presents the results in form of boxplots that better illustrated what we describe in the text. We also add a paragraph in the method section.

L 295/296 "A peak of high event water at the beginning of the sprinkling experiments,..." I could not find this result in the presented data (Fig. 3?).

We agree that this is difficult to see as the total SSF at the beginning of the event is small in general. We will upload an example plot that shows new water fraction as a function of time. However, the high content of pre-event water in SSF during the entire experiment suggests, that preferential flow is a secondary process.

L 302 Tab. 1 (skeleton content) and Fig. 1 (soil bulk density)

We will add/correct the cross-references

L 315-317 Why is the Ptot concentration from the mineral soil in vertical SSF in MIT lower than from the forest floor?

Probably your question aims at the fact that only in MIT the Ptot concentration in LATERAL SFF from mineral soil is higher than in the LATERAL SFF from the humus layer. A possible explanation is given in Line 317-320: "This is explained by the difference in P-stocks of the forest floor and mineral soil of the three sites. While Ptot stocks in the forest floor at MIT are only 7 g/m2 it is almost 2 times higher at CON (13 g/m2) and almost three times higher at TUT(19 g/m2) (see **Error! Reference source not** found.). On the contrary the Ptot stocks in the mineral soil at MIT (624 g/m2) are almost 3 times higher than at CON (230 g/m2) and more than three times higher than at TUT (189 g/m2)".

In addition lateral SSF from the forest floor at MIT was larger than lateral SSF from the mineral soil while this is not the case for CON and TUT (see Fig. 3b)."

L 339/340 This is only true for vertical SSF, isn't it?

Yes, we will add vertical SSF

L345 This is predominately the case for LY1B, isn't it (Suppl. Tab.1)?

Yes, we will delete the sentence in L345f and fit the information at the end of section 3.4.1.

L 350/351 Which soil properties?

**We add e.g., drainable porosity**

L 361 It is unlikely that adsorption explains the difference, since adsorption is very small in the forest floor. How large was the P flow from the 3 sites in g/m2 (in cormparison to the soil P stocks of the 3 sites)? Compare it with values from the literature that you cited in the introduction (L 30 and others).

We will rewrite this part to make clear that we think most of the lateral SSF from the forest floor is likely to occur at the contact face between the relatively high permeable forest floor and the lesser permeable mineral soil. So TR1B likely receives water that was flowing at or near the surface of the mineral soil towards the trench. Along this surface of the mineral soil, adsorption can happen. LY1B is installed below the forest floor but on top of the mineral soil and therefore collects water that did not have contact with the mineral soil.

Tab.2 Why was the soil depth of the installations in the subsoil in CON different from MIT?

The depth of installation was adapted to the depth of the soil horizons that differ between sites.

Tab. 4 and 5: You abbreviate both variables with trise20; better add "in SSF" in Tab. 4 and "in event water fraction" in Tab. 5.

**We will do this**

Fig. 3b Reorder the labels (TR1B, TR2B, TR3B)according to the labels in Fig. 3a (from forest floor to saprolite).

We will address this in the final version

Fig. 5 The unit of the flow on the x-axis is mm/h, isn' it?

Yes, we will change that. Thank you!

Technical corrections:

Thanks for pointing these issues out. We will address all technical corrections

- L 29 forest ecosystem -> forest ecosystems
- L 66 In biopores-> Biopores
- L 171 chemotatic -> chemostatic
- L 225 paranthesis is missing
- L 287suction caps -> suction cups
- L 332 Makoswski et al -> Makowski et al.
- L 338 suggest-> suggests
- L 345 was -> were
- L 357 and 370 expect -> except
- Tab. 1 Dominant vegetation and Annual precipitation for TUT: d -> c
- Fig. 5 and Fig. 6 Labels -> Label

---

## Author Response (AR1)

**Editor Comment**

*As you have seen, the reviewers are generally positive about your study. I also appreciate the vast amount of work that went into this elaborate field experiment that generated very interesting data. Reviewer 1 raised however an important point: you present water fluxes and P concentrations, but failed to calculate the obvious thing: P fluxes. In your answer you refer to another manuscript that you are working on, which specifically deals with P fluxes. For an editor, this is frankly not very satisfying, because how do you guarantee that the present publication is not obsolete, the moment the second will be published? Again, I appreciate the vast amount of work that went into this experiment and I can understand that you want to get the necessary impact out of it. On the other hand I am not a big fan of splitting data to get more publications. So, please reconsider whether it is smart to publish the P fluxes in a seperate paper, or provide good scientific arguments why is is worth published the concentrations and water fluxes separately from the P fluxes. In any case I expect that, independent of your decision, substantial changes will be necessary to your manuscript because at present a lot of the introduction is focussing of P fluxes even though they are not part of the results.*

*Best regards,*

*Edzo Veldkamp*

**Response to the Editor:**

Dear Prof. Veldkamp,

Thank you for handling the review process of our paper manuscript. In the revised version we just uploaded we have addressed every comment of both reviewers (see separate uploads). Here we summarize the most substantial changes:

As suggested, we have changed the focus of the paper manuscript and present our results on subsurface storm flow and P concentrations in the light of P turnover an P buffering capacity of soils. As the term "turnover" is commonly used in the context of biotic processes but our study analysis the net effect of both biotic and abiotic processes we decided to use the term *rate of P replenishment* as a more general and neutral term. This term and its meaning is explained in the beginning of the introduction section.

- We have changed the title in line of the suggestion by reviewer #1
- we have rewritten the abstract
- and a substantial part of the introduction
- have added information on P stocks and isotopically exchangeable P

- have added a whole new section in the discussion that reflects the change in focus of the paper.
- We have included a rough estimation of the annual P flux for CON which is the only site we have data for more than one year. We compare it to a study by Sohrt et all. (2019) who estimated annual P fluxes from be-weekly groundwater samples. We also compare our annual P fluxes with input fluxes by dry and wet deposition and rock weathering from this site and from other studies and conclude that P fluxes will be relevant for longer time scales as they are in a similar order than dry and wet depositions or mineral weathering.
- The conclusions are rewritten as suggested to less summarize the results but highlight the new findings.

With this substantial revision we think that we have adequately responded to the feedback and suggestions by both reviewers. They have stated that the content we presented in the first submission was interesting to the scientific community. Including the reviewers comments has certainly improved the paper but we aim to convince that presenting results from a field study like this is relevant to the scientific community. Results on subsurface flow and their interpretation in terms of runoff generation mechanisms and response time and old and new water is still missing in the field and is limiting advance in process understanding. There has been recently an explicit call for more field based experimental work in our field (Burt & McDonnell, 2015). Our study has a profound experimental setup that allowed to measure not only lateral but also vertical subsurface flow – something that has been neglected in previous studies. The experiments also improve on limitations of earlier (smaller-scale, shallower depth) studies that are (more) likely affected by soil heterogeneity and boundary effects.

We also think that our quantitative, hypothesis driven attempt to test whether subsurface flow in various soil depth is subject to dilution/enrichment or is chemostatic across a spectrum of soils with different properties is worth presenting and will not be obsolete in the future. As one of the reviewers mentions, a study like ours has implications on the land surface modelling community that seeks for simple but realistic (and therefore empirically tested) assumptions to model long-term nutrient cycling. To better illustrate the relevance of our outcome and its potential use we now include an example of a rough estimation of the annual P flux for one of our sites where we have the necessary data. We think that the revised paper manuscript and title is now more focused to our aim and offers convincing evidence that this is a relevant contribution to the scientific community.

Best regards,

**Response to the Reviewers:**

**Reviewer 1**

Anonymous Referee #1:
*We thank the reviewer1 for his/her assessment of our paper manuscript and the useful comments to improve the text. To make it more convenient for the second round of reviews we have provide this updated version of our original response to the reviewers. This way we hope to guarantee a better readability.*

General remark: The authors present data from sprinkling experiment in three forest sites, performed during two different seasons, where they analyzed water flow and soil solution P con-centrations. The paper is generally well-written and easy to follow, and the results are interesting.  However,  from my point of view the motivation/objective of the paper is not yet properly addressed with the results.  This needs to be addressed before the manuscript can be published. The stated objective of this paper to quantify P losses via subsurface flow (abstract, as well as l.  75 of introduction). This sets the reader up to expect to learn about phosphorus fluxes.  More information on subsurface flow P losses would indeed be very interesting, also for the land surface modelling community, which is struggling to incorporate P cycling into C, N models.  However, in the results no soil P fluxes [g Pm-2 time-1] are presented, only P concentrations [mg P L-1 water].  I suggest authors to bring the paper in line with the objectives. Firstly, in the introduction by introducing what are typical soil P stocks in forest ecosystems (see e.g. (Achat et al. 2016; Hou etal. 2018)), and further what are orders of magnitude for P flux losses (e.g. in g P m-2yr-1) as determined by earlier studies (see e.g. (Vitousek 2004) and others authors would have to search the literature a bit here).   Perhaps also comparing to other Pfluxes in forest ecosystems such as dust deposition, rock weathering, etc. This will set the scene for talking about P fluxes in forest ecosystems. I'm guessing that the losses will be several orders of magnitude lower than the stocks, and it will have to be argued why (if?) they are still important. Secondly, no P flux data is presented in the results. Is it possible to multiply water flowby P conc. to get P flux? Why is this not done?

*The focus of the paper has now changed and our aim is now better reflected by a new title.*

*One of our findings was that chemostatic transport conditions were prevalent in the mineral soil for most of the experiment, even for very high rainfall amounts. This would suggest that annual P flux from forest stands could be approximated by simply knowing the average P concentration and the water balance of the site. To illustrate this, we have now included a new section in the discussion where we estimated the annual P flux for CON. We compare our annual P flux with results from other studies and with P inputs by dry and wet deposition and weathering.*

The discussion should be developed further also. How do the results from this study tie into what we already know about P cycling in forests, and P loss pathways? At the moment the discussion mostly explains the results, but it needs to go further to show readers what has been learned. Again, given the setup of the paper, the focus should be on P fluxes. What do the results mean in terms of fluxes? What do we learn about P cycling in forest ecosystems?

*We have revised the discussion section to better match the new title. We focus a) on process understanding of subsurface flow (SSF) mechanisms and on b) the P-transport conditions during the experiments. We hope to now better discuss and explain why findings like presented in this paper are useful. We do this (among others) by presenting a rough estimate of an annual P flux that is based on the assumption of chemostatic transport conditions.*

Just thinking out loud (authors may choose to followup on this or not): Apart from the nutrient flush in the first 1-2 hours, P concentrations were relatively constant regardless of SSF. On a methodological note, does this imply that we can (roughly) approximate annual P losses via SSF given the water balance of the site and the soil solution P concentration? What would that imply in terms of annual P loss [g P m-2 yr-1] for these sites? How does that compare to the forest stocks and orders of magnitude that can be expected for other loss and input pathways such as dust deposition, weathering and erosion (Chadwick et al. 1999; Hartmann et al. 2014; Tipping et al. 2014; Aciego et al. 2017) ?

*We have now included a rough estimation of the annual P flux for CON which is the only site we have data for more than one year. We compare it to a study by Sohrt et all., (2019) at the same site who estimated annual P fluxes with a different approach. We also compare our annual P fluxes with input fluxes by dry and wet deposition and rock weathering from this site.*

Specific comments

Title: This is up to the authors, but if they want their article to also reach hydrologists, the title (and abstract?) should be revised. A good portion of the results and discussion as well as the conclusion focus on water flow, which I did not expect from reading the title. E.g. something along the lines o f "Beech forest stands sprinkling experiments: effects on sub-surface flow and phosphorus dynamics"

*We change the title in line with the suggestion and rewrote the abstract.*

l. 23 Jumping on the "climate change" bandwagon here is unwarranted. There is no discussion of climate change in the article. Also, the data rather show that P conc. is constant and thus only dependent on water balance, right?

125 *The idea of this sentence was to put the paper in a very broad, general context but we agree that climate change is not a main theme in the following paper analysis. Still, precipitation is predicted to change as a consequence of climate change, and by this will have an effect on SSF and thus P-transport. We have deleted this sentence.*

l. 29 How much P is in forest soils? How big are these losses?

130 *We have deleted this sentence as we focus on P dynamics*

l. 32 remove period after "SSF"

*We have rewritten this part*

l. 34 remove period after "nutrients"

*We have address that*

135

l. 45 The way this sentence is written makes it sound like it was done in this study. I suggest to change tense to "has been" or state "in previous studies"

*We have address that*

140 l. 52 add ", USA"

*We have address that*

l. 54-62 This is too detailed and should be condensed

*We have address that*

145

l. 66 "In biopores..." ?

*We have corrected this*

l. 74 "We performed....to capture potential differences in P fluxes." However, in the research questions the focus is on dynamics of P concentrations. This should be aligned.

150 *We have change it to P dynamics*

l. 99 231 g at CON is very similar to 209 g at TUT, especially given heterogeneity inherent to soils. I don't think you can argue that TUT is "less rich in soil P" than CON.

*We realize that the way we wrote the sentence is maybe misleading to the reader. We have stated the P-content but not rank it*

155 *relative to each other.*

l. 99 So that the reader can put these numbers into relation (is 209 – 678 g P m2 really a large range in P, justifying calling one P poor and the other P rich?), I again suggest presenting orders of magnitude ranges in soil P stocks in forests (see comment l. 29)

*We avoid the terms "rich" and "poor" in the paper and just state the numbers.*

l. 102 Add period before "Bulk". I stop correcting spelling / grammar mistakes at this point, but there are more in the remaining text. Please proof read the next version carefully.

*We have worked on that.*

l. 136 I'm no expert here, but I'm guessing rain water is far from de-ionized. How do you think using deionized water affected the results? Does that need to be discussed?

*Collecting 60.000 L of rainfall for the experiment was not an option. So, we were left with using groundwater from the drinking supply system. We argue that using untreated groundwater as sprinkling water would have been unacceptable from an experimental design point of view simply because it is an unnatural source of hydrochemical compounds (including P) to the system. We think that the term "deionized" might make some readers think of purified water like in a lab environment. To show that this was not the case, we had added in L 136 (first submission) that the water had an electrical conductivity of 20 μS/cm. This is comparable to the EC of natural rainfall. The 20 μS/cm is a result of the efficiency of the industrial deionizer and processing 60.000 L of water. In order to avoid irritation, we have removed the term "deionized water".*

l. 170-177 Nice setup to let the reader now what to expect, look for and interpret in the results! That's an example of great scientific writing .

*We appreciate your positive comment*

Table 1: Please also add pH to the table. pH is an important indicator of soil P forms and dynamics and may be important to explain the results, e.g. the difference between TUT, CON and MIT.

*We have added pH to Tab 1*

Fig. 1 and others: colors are not grayscale print-friendly:

*We argue to keep figures in color-scheme as showing all in gray scale is hard to indicate the information included in the plots.*

Fig. 2 Very nice overview figure. This makes it a lot easier to understand what was done.

*We appreciate your positive comment*

Fig 3. also this is a nice figure. I suggest to move spring before summer. I understand that spring experiments were carried out a year later, and that's ok since you have the dates there and it can be noted in the figure caption. But it makes more sense to have the plots in seasonal order for interpreting the plots

*We have changing the order of the panels.*

Section 3.4 It would have been interesting to measure inorganic and organic P as opposed to only total P.

*We see the reviewer's point but Ptot is the data that we have at hand.*

190 Results section 3.5: multiypling conc. by water flow = element flux. Why not present these data in a section 3.5 "Soil P fluxes"

*The focus of the paper has changes to better reflect our aim to study SSF and P -dynamics. The fact that we observed chemostatic conditions during most of the duration of the experiment is a useful information for the community. To illustrate our point we have includes a rough estimation of the annual P flux based on SSF data and chemostatic transport conditions;*
195 *an assumptions motivated by parts of our results.*

l. 256-260 (p. 8-9): I'm not surprised that P conc. in the soil solution remains relatively constant. If we consider the very fast turnover time of P in the soil solution of only seconds to minutes (Helfenstein et al. 2018).

*We thank the reviewer for the literature suggestions. We have now included turnover (or more general rate of replenishment) and buffering in the introduction and the discussion section (4.2).*

200 l. 300 What about biopores? Is there evidence to suggest that CON and TUT have more earthworms or other large soil fauna

*Biopores can also make a contribution to preferential flow. We have included this now in the text. Due to the low pH, there are no or only a few earthworms to be found at MIT and CON. In TUT, earthworms are present. However, due to the clay content, we consider it very likely that cracks originating from shrinking and swelling processes make the largest contribution to preferential flow.*

205

section 4.2 This section could be re-written to make it more focused. At the moment there is a mix of rather trivial findings, such as that P stocks are higher in the forest floor than in the mineral soil, while the interesting things are not discussed in-depth enough. The discussion of P concentration dynamics should be better linked to existing literature, e.g. what is known about turnover time of P in the soil solution and phosphate buffering capacity. Phosphorus-buffering capacity (PBC) is defined
210 as the ability of soil to moderate changes in the concentration of soil solution P (Beckett and White1964; Olsen and Khasawneh 1980; Barrow 1983; Pypers et al. 2006), and would be interesting to bring in here. Soil solution P turnover, a related concept, has been shown to be negatively correlated with P conc. in the soil solution (Helfenstein et al. 2018),which authors might consider discussing as well. (i.e. the more P in the soil solution(forest floor), the slower the turnover time; the less P in the soil solution (mineral soil),the faster the turnover time.

215 *Thank you for your valuable input. We agree that including the mentioned issues have improve the discussion. Section 4.2. has undergone major revisions along the lines suggested.*

l. 347 As with the plots, I would take spring before summer.

*We have changed this for all figures*

l. 364 not exactly true that you have six different experiments. It's one experiment carried out on three sites and at two time
220 points.

*We have change this.*

l. 371 It's quite well known that soil solution P concentrations are lower with increasing soil depth. I would rather focus on novel findings in the conclusion.

We have rewritten the text.

225 l. 372 "it was especially strong..." What is it?l. 373 It is obvious that P concentrations are highest in the P-rich site. Again, the conclusion should focus on the novel findings.

*We have rewritten the text.*

l. 374 "Particulalry high". Please be concrete. How much higher? Are we talking 1.5x,2x or 10x higher than during the rest of the experiment?

230 *We have added the actual value*

l. 375 – 379 This is interesting and in my opinion the main finding of the study. This should be placed more prominently and discussed appropriately.

*We have extended this part*

l. 380 Conclusion not supported by the data. There was no discussion of climatechange in the article.

235 *The last sentence of the paper has been removed.*

l. 436 "DWD, 2010" please provide complete citation reference

Reference is updated

**Reviewer 2**

Anonymous Referee #2:

General comments: The manuscript entitled " Phosphorus Transport in Subsurface Flowat Beech Forest Stands:  Does Phosphorus Mobilization Keep up with Transport? ",written by Michael Rinderer, Jaane Krüger, Friederike Lang, Heike Puhlmann,  and Markus Weiler, presents valuable results that contribute to the understanding of phosphorus transport in and phosphorus losses from the soil. The topic falls into the scope of Biogeosciences. The manuscript comprises results from  large sprinkling  experiments at three beech forest sites in Germany.  The methods are adequate to test the research questions.  The results are described in detail and can be used to answer the research question.  The text is easily understandable, tables and figures are well-arranged and the conclusions are sound. Hence, I would recommend to consider this manuscript for publication in Biogeosciences after minor revision.

*We thank the reviewer2 for his/her positive assessment of our paper manuscript and the useful comments to improve the text. In the following we respond to each comment. Our response is similar to the one we had uploaded earlier.*

Specific comments

L 14 The values differ from those in Tab.  1.

*We corrected the values to match Table 1.1*

L75/76 The time of the two experiments was not well chosen if microbial conditions – like soil moisture, temperature, litter fall – should differ. Rather late autumn/early winter (november; wet, cold, a lot of litter) and summer (july/august; dry, warm, less litter) should have been chosen.

*We agree that a stronger contrast in seasonality would have been better to evaluate seasonal effects. However, this was a subordinate part of the study and is therefore not listed as a separate research question in our paper. When choosing our days of sprinkling we were restricted to the vegetation period (i.e., the time when trees had leaves and active photosynthesis) as we were also monitoring tree water uptake and P-transport in trees during the subsequent 4 to 6 weeks after the sprinkling experiment.*

*However, we will rewrite the text as follows:*

*"We performed two sprinkling experiments at each site to capture potential differences in P fluxes within the vegetation period (i.e., between summer/fall and spring). …"*

*And we deleted the part "…and litter fall is not evenly distributed across the year." from the manuscript.*

L 227 trise20 of the event water fraction is in Tab. 5 and trise20 of SSF in Tab.4

*Thanks for pointing this out. We corrected it.*

L 233-252 Results of the statistical analyses are not displayed anywhere and the statistical approach is not described in the materials and methods section.

*In addition to Figure 4 we have added another figure (Figure 5) that presents the results in form of boxplots that better illustrated what we describe in the text. We also add information in 2.3 Data Analysis (second paragraph).*

L 295/296 "A peak of high event water at the beginning of the sprinkling experiments,..." I could not find this result in the presented data (Fig. 3?).

*We agree that this is difficult to see as the total SSF at the beginning of the event is small in general. Here is an example that shows high new water fraction at the onset of response. However, cases like this were few. In general, the high fraction of pre-event water in SSF during the experiment suggests, that preferential flow is a secondary process.*

[Figure]

L 302 Tab. 1 (skeleton content) and Fig. 1 (soil bulk density)

*We will add/correct the cross-references.*

L 315-317 Why is the Ptot concentration from the mineral soil in vertical SSF in MIT lower than from the forest floor?

Probably your question aims at the fact that only in MIT the Ptot concentration in LATERAL SFF from mineral soil is higher than in the LATERAL SFF from the humus layer. A possible explanation was given in Line 317-320 of the first submission version of the paper: "*This is explained by the difference in P-stocks of the forest floor and mineral soil of the three sites. While Ptot stocks in the forest floor at MIT are only 7 $g/m^2$ it is almost 2 times higher at CON (13 $g/m^2$) and almost three times higher at TUT(19 $g/m^2$). On the contrary the Ptot stocks in the mineral soil at MIT (624 $g/m^2$) are almost 3 times higher than at CON (230 $g/m^2$) and more than three times higher than at TUT (189 $g/m^2$)*".

In addition, lateral SSF from the forest floor at MIT was larger than lateral SSF from the mineral soil while this is not the case for CON and TUT (see Fig. 3b)."

Reviewer #1 suggested to rewrite the section 4.2 to better match the new focus and outcome of the paper. Therefore, the sentence is no longer part of the paper.

L 339/340 This is only true for vertical SSF, isn't it?

*Section 4.2. has undergone major rewriting and the original sentence has been changed.*

L345 This is predominately the case for LY1B, isn't it (Suppl. Tab.1)?

Yes, we will delete the sentence in L345f and fit the information at the end of section 3.4.1.

325    L 350/351 Which soil properties?

We                 add               e.g.                ,              drainable             porosity

L 361 It is unlikely that adsorption explains the difference, since adsorption is very small in the forest floor. How large was the P flow from the 3 sites in g/m2 (in cormparison to the soil P stocks of the 3 sites)? Compare it with values from the literature that you cited in the introduction (L 30 and others).

330    In the course of rewriting section 4.2. this part has gone but we think that most of the lateral SSF from the forest floor is likely to occur at the contact face between the relatively high permeable forest floor and the lesser permeable mineral soil. So TR1B likely receives water that was flowing at or near the surface of the mineral soil towards the trench. Along this surface of the mineral soil, adsorption can happen. LY1B is installed below the forest floor but on top of the mineral soil and therefore collects water that did not have contact with the mineral soil.

335    Tab.2 Why was the soil depth of the installations in the subsoil in CON different from MIT?

The depth of installation was adapted to the depth of the soil horizons that differ between sites.

Tab. 4 and 5: You abbreviate both variables with trise20; better add "in SSF" in Tab. 4 and "in event water fraction" in Tab. 5.

We have deleted the abbreviation in Table 4 and 5.

340    Fig. 3b Reorder the labels (TR1B, TR2B, TR3B)according to the labels in Fig. 3a (from forest floor to saprolite).

We have change this.

Fig. 5 The unit of the flow on the x-axis is mm/h, isn' it?

Yes, have change that. Thank you!

Technical corrections:

345    Thanks for pointing these issues out. We have address all technical corrections.

L 29 forest ecosystem -> forest ecosystems

L 66 In biopores-> Biopores

L 171 chemotatic -> chemostatic

L 225 paranthesis is missing

350    L 287suction caps -> suction cups

L 332 Makoswski et al -> Makowski et al.

L 338 suggest-> suggests

L 345 was -> were

L 357 and 370 expect -> except

355    Tab. 1 Dominant vegetation and Annual precipitation for TUT: d -> c

Fig. 5 and Fig. 6 Labels -> Label

**Manuscript with tracked changes:**

[revised manuscript text omitted]

---

## Referee Report (RR1)

**General Comments**

The manuscript entitled „Subsurface Flow and Phosphorus Dynamics in Beech Forest Hillslopes during Sprinkling Experiments: How fast is Phosphorus replenished?" written by Michael Rinderer, Jaane Krüger, Friederike Lang, Heike Puhlmann, and Markus Weiler was rewritten according to the suggestions of the editor and two referees. The main concern was that P fluxes were not presented, even though one of the objectives was to quantify P losses with subsurface flow. The authors substantially rewrote the manuscript. They focus now on soil solution P replenishment and P buffering capacity of the mineral soil as these are important processes for plant P nutrition. In addition, they make a rough estimation of the annual P flux from one of their research sites.

Unfortunately, I have more concerns with the new version of the manuscript than with the first one. These are the reasons:

1. Now the authors try to meet two goals, which makes the manuscript more difficult to read. They start the introduction with plant P nutrition issues, but do not come back to them in the discussion. They should for example discuss the implications of P leaching from the forest floor and P retention in the mineral soil for plant nutrition. What about the effect of different tree species? They cite several studies form other beech forest ecosystems, but are there also results from other tree species? Nutrient flushing might be of great importance for plant nutrition. What are the processes that lead to P flushing? What role do microorganisms play?

2. In order to be able to estimate the annual P flux of one of the three research sites, the authors used the P concentration after the nutrient flush and ignored the high soil solution P concentrations during the first two hours of the experiment (nutrient flush). In the first version of the manuscript the authors wrote that climate change may lead to more frequent high-intense rainfall events. Hence, nutrient flushing might become more relevant and it would be interesting how large the P flux would be during such high-intense rainfall events (not annually). In addition, the authors write in line 400 that "up to 40 % of the annual P flux might occur during single events, which would suggest that the fluxes during the first flush have an important share on the annual flux and cannot be neglected". The P flux during the sprinkling experiment (first flush and rest of the time) could have been calculated for all three research sites.

Hence, I would suggest to either focus on P replenishment and buffering or P fluxes from the ecosystem and to discuss the chosen topic in more detail.

**Specific Comments**

L 170-190 Still, the statistical methods used are not described. In line 247 you write "was significantly higher", but it is not explained anywhere how you tested this.

L 205 The trench cannot yield vertical SSF.

L 297 You found that especially vertical SSF is of importance for P transport. Hence, suction cups installed in a large number and spread over a research site, should yield good estimates of soil solution P concentrations, especially when soil heterogeneity is high.

L 336-339 Explain the process behind nutrient flushing.

L 367-368 This is only right if the nutrient flush at the beginning of the experiment is not relevant. You did not calculate the flux for the flush, hence, I cannot confirm that your statement is right.

L 397 Which ecosystems are included in the meta-analysis of Sohrt et al. (2019)?

**Technical corrections**

L 243 6 time -> 6 times

L 398 either delete the colon or the parentheses

L 419 suggest -> suggests

L 431 ecosystem -> ecosystems

L 432 relative -> relatively

---

## Author Response (AR2)

**Response to the Editor:**

Dear Prof. Veldkamp,

We thank you and the two reviewers for the useful feedback to the revised version of our paper. We have addressed the comments and have uploaded an updated version.

In the following we respond to the individual comments (in red color).

Best regards,

10   Michael Rinderer and co-authors

**Editor Comment**

15   *Dear Dr. Rinderer and colleagues,*

*I have now read your revised manuscript and the reviews provided by the two referees.*

*You will also see that the referees made some contrasting recommendations based on your revised manuscript. Referee #2 is very positive about the changes in the direction of the manuscript, Referee #1 expresses some concerns, although this referee*

20   *continues to be very positive about the overall scientific significance and quality of your study.*

*I would like to commend you for taking the reviews seriously and making considerable changes to your original manuscript based on the earlier reviews. I think that the manuscript has improved substantially, both in focus and the message it conveys.*

*Nevertheless, I would recommend some additional changes which I would consider relatively minor:*

25   *-Referee 1 expresses concern that the importance of nutrient flushing is not sufficiently stressed in your discussion. My impression from your results is that the nutrient flushing was not contributing enough (both in terms of waterflow and P concentration) to have a significant effect on the annual P losses from your ecosystem and so your approach to calculate annual P fluxes appeared reasonable. However, referee #1 makes a good point when referring to line 400, in which you cite another study which claims that up to 40% of the annual P flux may occur during single events. Does this mean that your study*

30   *results deviate from this earlier study? To address the concerns of Referee #1, I suggest to include in your discussion a short section in which you address what the potential error is when nutrient flushing during single events is ignored in your annual budget.*

We are aware that the P losses during the flushing phase are neglected in this simple calculation of the annual P losses. To be clear that this part of the paper is a hypothetical test based on assumptions we now write at the beginning of section 4.3: *"If one would argue that the amount of SSF during the flushing period is small compared to the remaining part of the event (see Fig. 3) one could also assume that chemotactic conditions prevail during large rainfall events. Given these simplifications, one can roughly estimate the annual P losses by knowing the amount of annual SSF and an average P concentration."*
We have deliberately cited the study by Julich (personal communication cit. in Boll et al. (2016)) to show that we are self-critical about these simplifications and consider these as potential reasons why some other studies cited in a meta-analysis by Sohrt et al. (2017) show higher values.

And at the end of the second paragraph of section 4.3 Line 419 we write: "Given the high P concentrations during the flushing phase of our sprinkling events it might be that relatively small amounts of SSF during the first flush can still export a significant amount of P."

*-Referee #1 also suggests to include the effect of different tree species. Because your study sites were all dominated by beech trees I do not see that your study can contribute much new information to this interesting question. As your manuscript is already relatively long, I suggest not to explore this further in your present manuscript.*
We agree that studying differences between tree species is an interesting topic but as you said, we do not have the data to contribute to it.

*Both referees have also also added a list of specific comments, which I would like you to address. Especially the statistics should be described if you conducted them.*
We have addressed each comment by reviewer 1 and 2. In terms of the statistics we added the information requested by reviewer 2 in the Method section and repeated this information later where the reviewer asked us to do.

*A few additional suggestions:*
*-In your response you write about 'quantitative, hypothesis-driven ... test' (l.49). Unfortunately, you did not write out your hypotheses in the introduction, which I strongly recommend.*
We have added hypothesis at the end of the introduction section.

*-Fig.1: I suggest to add error bars to the measured bulkdensity data and to connect the bulkdensity data with straight lines (unless you have a good scientific reason why you used non-linear interpolation).*
We added error bats and changed the type of line connectors.

*-L. 250: I suggest to use 'detection limit' instead of 'limit of quantification'. 'Detection limit' is not only used for analyses done by analytical equipment but can also include data processing steps.*

We distinguish between the limit of quantitation for Ptot (in our case 0.009 mg/l,) and the limit of detection (in our case 0.004 mg/l) based on DIN 32645 with a significance bound of 99 % for the limit of quantitation and 77 % for the limit of detection (DIN, 2008). Therefore, we need to keep the expression in figures and tables.

70

*I look forward to your revisions,*

*Best regards,*

75

*Edzo Veldkamp*

Thank you for handling our review process

80  Fond regards

Michael Rinderer and co-authors

**Response to the Reviewers:**

**Reviewer 1**

Anonymous Referee #1:

*The performed revisions improve the quality and usefulness of the manuscript. From my point of view the most interesting finding is that observed chemostatic transport conditions suggest that annual P losses at the lateral and vertical boundary of a forest plot can be approximated by knowing the average P concentration and the water fluxes in forest soils. The authors now point this out nicely. Other findings, such as that annual P losses are negligible compared to P stocks are not as interesting since it is well known that P is much more retained in ecosystems than other nutrients. But I think the manuscript is ready to be published now. Just some minor corrections below.*

We thank reviewer 1 for his/her positive feedback and suggestions that we have included in the new version of the manuscript. Despite the fact that our rough estimation of the annual P losses from the forest ecosystem based on subsurface flow data and an average P concentration we want to emphasize that work still needs to be done to compare the P losses during the flushing period with the losses during the remaining part of an event.

*Minor comments*

*l. 14. Rates do not have the unit "time" but the unit "time-1". If you are talking about replenishment times please use correct wording*

Indeed; we have replaced "rates" with "times"

*l. 28 "be-weekly" change to bi-weekly*

Thanks for pointing it out.

*l. 38 Again, turnover rate has the unit time-1, so please call it turnover time if you are talking about time*

We have replaced "rates" with "times"

*l. 327 Remove comma before "why"*

We have removed the comma

*l. 333 "Dynamics" should not be capitalized*

Originally, we followed the rule to capitalize all nouns in headlines. We have now changed all headlines including the title to have no capitalization except for the first word.

*l. 361 This sentence is strange. Since the experiment was in spring (beginning of vegetation period) and summer (full vegetation) I don't understand why you can say that the results confirm "higher P concentrations of leachate from the forest floor during the vegetation period". As far as I know there were no measurements in winter in this experiment*

We agree that we have no measurements during winter and therefore cannot strictly compare our results with other work.

*l. 364 change "form" to "from"*

Thanks for pointing it out!

*l. 426 average P concentration is vague. Please specify concentration in what. Soil solution? Leachate?*

We added soil solution to be clear.

*l. 427 Delete "quick"*

We have deleted quick

*l. 429 Add a comma after "budget"*

Thanks for pointing it out!

*l. 430 Replace first than with as*

Thanks for pointing it out!

**Reviewer 2**

Anonymous Referee #2:

*General comments*

140   *The manuscript entitled „Subsurface Flow and Phosphorus Dynamics in Beech Forest Hillslopes during Sprinkling Experiments: How fast is Phosphorus replenished?" written by Michael Rinderer, Jaane Krüger, Friederike Lang, Heike Puhlmann, and Markus Weiler was rewritten according to the suggestions of the editor and two referees. The main concern was that P fluxes were not presented, even though one of the objectives was to quantify P losses with subsurface flow. The authors substantially rewrote the manuscript. They focus now on soil solution P replenishment and P buffering capacity of the*

145   *mineral soil as these are important processes for plant P nutrition. In addition, they make a rough estimation of the annual P flux from one of their research sites.*

*Unfortunately, I have more concerns with the new version of the manuscript than with the first one. These are the reasons:*

*1. Now the authors try to meet two goals, which makes the manuscript more difficult to read. They start the introduction with plant P nutrition issues, but do not come back to them in the discussion. They should for example discuss the implications of*

150   *P leaching from the forest floor and P retention in the mineral soil for plant nutrition. What about the effect of different tree species? They cite several studies form other beech forest ecosystems, but are there also results from other tree species? Nutrient flushing might be of great importance for plant nutrition. What are the processes that lead to P flushing? What role do microorganisms play?*

155   We thank the reviewer 2 for his/her useful comments. We have included a text in the discussion section, line 421 that discusses implications of P leaching from the forest floor and P retention in the mineral soil for plant nutrition. We also end the abstract and the collusion section with this thought.

We agree that a similar study would be interesting in forests dominated by other species but as the Editor mentioned, we have only worked in beech forest stands and have no data to contribute to this interesting toic. We therefore found it more useful to

160   compare our results with other studies in beech forests.

*2. In order to be able to estimate the annual P flux of one of the three research sites, the authors used the P concentration after the nutrient flush and ignored the high soil solution P concentrations during the first two hours of the experiment (nutrient flush). In the first version of the manuscript the authors wrote that climate change may lead to more frequent high-intense*

165   *rainfall events. Hence, nutrient flushing might become more relevant and it would be interesting how large the P flux would be during such high-intense rainfall events (not annually). In addition, the authors write in line 400 that "up to 40 % of the annual P flux might occur during single events, which would suggest that the fluxes during the first flush have an important*

*share on the annual flux and cannot be neglected". The P flux during the sprinkling experiment (first flush and rest of the time) could have been calculated for all three research sites.*

*Hence, I would suggest to either focus on P replenishment and buffering or P fluxes from the ecosystem and to discuss the chosen topic in more detail.*

We are aware that the P losses during the flushing phase are neglected in this simple calculation of the annual P losses. To be clear that this part of the paper is a hypothetical test based on assumptions we now write at the beginning of section 4.3: *"If one would argue that the amount of SSF during the flushing period is small compared to the remaining part of the event (see Fig. 3) one could also assume that chemotactic conditions prevail during large rainfall events. Given these simplifications, one can roughly estimate the annual P losses by knowing the amount of annual SSF and an average P concentration."*
We have deliberately cited the study by Julich (personal communication cit. in Boll et al. (2016)) to show that we are self-critical about these simplifications and consider these as potential reasons why some other studies cited in a meta-analysis by Sohrt et al. (2017) show higher values.

And at the end of the second paragraph of section 4.3, line 419 we write: "Given the high P concentrations during the flushing phase of our sprinkling events it might be that relatively small amounts of SSF during the first flush can still export a significant amount of P."

Mentioning P replenishment/buffering and P fluxes in one paper is a result of feedback during the first round of reviews.

*Specific comments*
*L 170-190 Still, the statistical methods used are not described. In line 247 you write "was significantly higher", but it is not explained anywhere how you tested this.*
We have added the used statistical tests in section 2.3. To be very clear we repeated information on statistical tests in line 259.

*L 205 The trench cannot yield vertical SSF.*
Yes, this was a mistake. We have corrected this.

*L 297 You found that especially vertical SSF is of importance for P transport. Hence, suction cups installed in a large number and spread over a research site, should yield good estimates of soil solution P concentrations, especially when soil heterogeneity is high.*

We agree that suction caps can yield interesting information on soil solution P. They have advantages in terms of spatial coverage. As suction cups cannot be used to measure subsurface storm flow, we think that the large lysimeter plates were a good choice for the purpose of our study.

205 *L 336-339 Explain the process behind nutrient flushing.*

We have included a paragraph in section 42, line 353ff about processes leading to hight P concentrations during the onset of SSF.

*L 367-368 This is only right if the nutrient flush at the beginning of the experiment is not relevant. You did not calculate the*
210 *flux for the flush, hence, I cannot confirm that your statement is right.*

Yes, agreed. We have chosen a misleading way of writing. It was not meant to leave the impression we deliberately neglect the flushing phase. We think that the old text of section 4.3 had mentioned that by saying *"A possible reason for the higher values compared to our results could be that we used mean P concentration measured after the first flush which results in rather conservative estimates."* Nevertheless, we now have improved the beginning of the section. We now write:

215 *"If one would argue that the amount of SSF during the flushing period is small compared to the remaining part of the event (see Fig. 3) one could also assume that chemotactic conditions prevail during large rainfall events. Given these simplifications, one could roughly estimate annual P losses from forest stands by knowing the amount of annual SSF and an average P concentration. (…)".*

220 And at the end of the second paragraph of section 4.3 we now additionally write: *(…) Further research is needed to clarify the role of P-flushing on the total P loss from forest ecosystems. Given the high P concentrations during the flushing phase of our sprinkling events it might be that relatively small amounts of SSF during the first flush can still export a relatively large amount of P."*

225 *L 397 Which ecosystems are included in the meta-analysis of Sohrt et al. (2019)?*

It is a worldwide compilation of data of studies in mainly cold temperate and boreal forest ecosystems.

*Technical corrections*
*L 243 6 time -> 6 times*
230 Thanks for pointing this out

*L 398 either delete the colon or the parentheses*
Thanks for pointing this out

235     *L 419 suggest -> suggests*
        Thanks for pointing this out

        *L 431 ecosystem -> ecosystems*
        Thanks for pointing this out
240
        *L 432 relative -> relatively*
        Thanks for pointing this out

[revised manuscript text omitted]